# Contractive Dynamical Imitation Policies for Efficient Out-of-Sample Recovery

**Amin Abyaneh**[1]*, **Mahrokh G. Boroujeni**[2]*, **Hsiu-Chin Lin**[1], **Giancarlo Ferrari-Trecate**[2]

[1] McGill University,[2] École Polytechnique Fédérale de Lausanne (EPFL)

## Abstract

Imitation learning is a data-driven approach to learning policies from expert behavior, but it is prone to unreliable outcomes in out-of-sample (OOS) regions. While previous research relying on stable dynamical systems guarantees convergence to a desired state, it often overlooks transient behavior. We propose a framework for learning policies modeled by contractive dynamical systems, ensuring that all policy rollouts converge regardless of perturbations, and in turn, enable efficient OOS recovery. By leveraging recurrent equilibrium networks and coupling layers, the policy structure guarantees contractivity for any parameter choice, which facilitates unconstrained optimization. We also provide theoretical upper bounds for worst-case and expected loss to rigorously establish the reliability of our method in deployment. Empirically, we demonstrate substantial OOS performance improvements for simulated robotic manipulation and navigation tasks. See sites.google.com/view/contractive-dynamical-policies for our codebase and highlight of the results.

## 1 Introduction

Imitation learning provides an intuitive policy optimization framework for executing complex robotic tasks by learning from expert demonstrations (Hussein et al., 2017). However, naively replicating expert behavior can raise safety concerns during deployment, as the robot's trajectory may become *unreliable* when facing out-of-sample (OOS) states. For instance, if the robot begins from an unknown initial state or encounters environmental perturbations, it could fail to maintain a consistent and reliable behavior. To address this issue, *the policy can be modeled using a dynamical system (DS)* with rigorous reliability certificates (Ravichandar et al., 2020; Dawson et al., 2023).

One such certificate is asymptotic stability (Devaney, 2021), which ensures that all trajectories eventually converge to the same equilibrium state, regardless of the initial condition or perturbations (Khansari-Zadeh & Billard, 2011; Rana et al., 2020; Abyaneh et al., 2024). While asymptotic stability guarantees convergence, it overlooks the transient behavior (Tsukamoto et al., 2021). This phenomenon is illustrated in Fig. 1 for an arbitrary OOS initial state, where the trajectory induced by the stable policy converges to the target but fails to imitate the expert. In contrast, *contractive* DS policies ensure that the resulting trajectories exponentially converge to each other, and ultimately to the equilibrium state (Ravichandar et al., 2017; Mohammadi et al., 2024). We leverage the stronger notion of contractivity (Lohmiller & Slotine, 1998) to ensure reliability in the transient phase, and as a result, enhance the imitation quality.

There are two main approaches to learning contractive DSs. The most common method relies on constrained

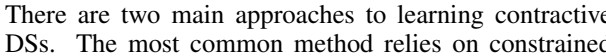

Figure 1: Policy rollouts generated by contractive and stable policies. While both policies eventually reach the target, the contractive policy closely mimics the expert in the transient phase.

---

*The first two authors collaborated equally.
Corresponding author: amin.abyaneh@mail.mcgill.ca

optimization to train the DS while satisfying learned or fixed contractivity certificates (Ravichandar et al., 2017; Blocher et al., 2017; Sindhwani et al., 2018). However, constrained optimization is computationally challenging for complex expert behaviors or high-dimensional state spaces. Additionally, most constrained approaches trade off imitation accuracy for constraint satisfaction. The second approach is based on parameterized models with built-in contractivity guarantees for any choice of parameters (Mohammadi et al., 2024; Dawson et al., 2023). As a result, the model remains contractive even if the learning process is imperfect or disrupted.

In this work, we build on the foundation of two such formulations: (i) a parameterization of contractive continuous-time DSs using recurrent equilibrium networks (RENs) (Martinelli et al., 2023), and (ii) the coupling layer architecture (Papamakarios et al., 2021). The former ensures that the DS is contractive with an adjustable rate, while the latter provides a trainable bijective transformation that preserves contraction properties. Our experiments demonstrate that the coupling layers enhance the model's representation power, leading to improved imitation accuracy and faster training, while maintaining the contractive nature of the DS.

**Related work.** Contractive imitation learning approaches are less common in the literature, primarily due to the increased difficulty of enforcing contractivity compared to asymptotic stability. The majority of existing methods train the DS under contractivity constraints (Blocher et al., 2017; Ravichandar et al., 2017; Sindhwani et al., 2018), resulting in a computationally demanding optimization problem. These methods assume specific representations of the DS or the contraction certificate to mitigate the intensive computation. For example, Ravichandar et al. (2017) employ sum-of-squares programming with polynomials, while Blocher et al. (2017) and Ravichandar et al. (2017) use Gaussian mixture techniques. Additionally, Sindhwani et al. (2018) adopt reproducing kernel Hilbert spaces. By focusing on these particular representations, these methods reduce the computational cost, but at the cost of limiting the model's representation power.

More recently, Mohammadi et al. (2024) introduce a promising unconstrained neural architecture with inherent contractivity guarantees. The algorithm optimizes a parameterized negative-definite Jacobian in a lower-dimensional latent space and integrates it to derive the DS. Despite better performance in some high-dimensional environments, it under-performs in lower-dimensional ones. The method also requires solving a second-order differential equation, which is more complex than the standard first-order equations typically used (Rana et al., 2020; Sochopoulos et al., 2024). Further discussion of the literature, including non-contractive methods, is provided in App. B.

**Contributions.** We propose a **S**tate-only framework for learning **C**ontractive **D**ynamical **S**ystem policies (SCDS), which offers several distinct advantages. First, SCDS learns the policy solely from *state measurements*, eliminating the need to measure the expert's velocity. Hence, we address the cumulative error problem observed in previous methods that replicate expert velocity, where small errors can accumulate and lead to significant deviations in state space trajectories (Ravichandar et al., 2020; Abyaneh et al., 2024). Additionally, SCDS enables learning in a *latent space*, with its dimension adapting to the problem at hand. It facilitates learning in high-dimensional state spaces by mapping to lower-dimensional latent representations, while also providing flexibility in low-dimensional spaces through higher-dimensional embeddings.

From a theoretical standpoint, we provide a rigorous solution for OOS recovery by establishing an *upper bound* on the worst-case deviation from expert trajectories. SCDS also offers control over the transient behavior through an *adjustable* and *learnable* contraction rate. Training SCDS is highly efficient by relying on unconstrained optimization and neural ordinary differential equations (ODE) fixed-point differentiation for gradient computation (Chen et al., 2018). An overview of SCDS is outlined in Fig. 2.

## 2 CONTRACTIVE POLICY FORMULATION

### 2.1 PROBLEM SETUP

**Dataset.** Consider a robot with a state $\mathbf{y}(t) \in \mathcal{Y}$ at time $t \in \mathbb{R}_{\geq 0}$, operating within a state space $\mathcal{Y} \subset \mathbb{R}^{N_y}$. The state $\mathbf{y}$ is typically considered to be the robot's joint or task space configuration.

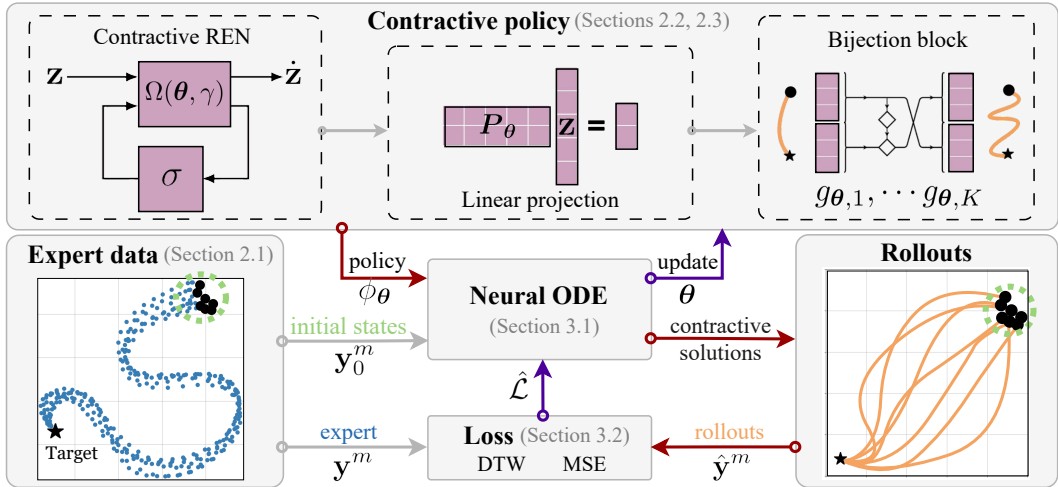

Figure 2: Overview of the SCDS training scheme. The policy structure (top box) consists of a REN, a linear projection, and a bijection block, ensuring contractivity for any choice of parameters, $\boldsymbol{\theta}$. In the forward pass ($\rightarrow$), initial states are passed through a differentiable ODE solver to generate state rollouts. The loss function penalizes the discrepancy between the generated and expert trajectories, updating the policy via backpropagation ($\leftarrow$).

Within $\mathcal{Y}$, we have a dataset $\mathcal{D}$ of $M \in \mathbb{N}$ expert demonstrations, defined as:

$$\mathcal{D} \triangleq \left\{ \mathbf{y}^m = \left( \mathbf{y}_0^m, \cdots, \mathbf{y}_{N_m-1}^m \right) \right\}_{m=1}^M, \tag{1}$$

where the $m$-th demonstration, $\mathbf{y}^m$, has $N_m$ samples. We assume that all expert trajectories converge to a common target state, $\mathbf{y}_{N_m-1}^m = \mathbf{y}^*$ for all $m$, but allow for different initial states and different sample sizes per trajectory. Note that our approach only requires state measurements, unlike most previous work, which rely on both the state and its time derivative.

**Objective.** We aim to learn a policy that precisely replicates the expert trajectories in the dataset $\mathcal{D}$ while ensuring robustness against perturbations during deployment. In other words, the policy should be able to recover safely if the robot encounters OOS states, either as a result of perturbations or different initial states from those in $\mathcal{D}$. We provide such guarantees by formulating the policy as a parameterized DS and leveraging tools from the contraction theory laid out in the next section.

## 2.2 BACKGROUND: CONTRACTIVE LATENT DYNAMICS

Consider an autonomous DS in continuous time $\dot{\mathbf{z}}(t) = f(\mathbf{z}(t))$, where $\mathbf{z} \in \mathbb{R}^{N_z}$ is the state and $f : \mathbb{R}^{N_z} \to \mathbb{R}^{N_z}$ specifies the dynamics. Contractivity is a fundamental characteristic of $f$, ensuring that the solution trajectories of $\mathbf{z}$ incrementally converge to a single trajectory, even when starting from different initial conditions or facing perturbations along the way (Lohmiller & Slotine, 1998; Tsukamoto et al., 2021). Therefore, a contractive DS provides stability and predictability as the state evolves over time. A formal definition is outlined in the following.

**Definition 2.1.** *(Lohmiller & Slotine, 1998) The dynamical system $\dot{\mathbf{z}}(t) = f(\mathbf{z}(t))$ is contracting with the rate $\gamma \in \mathbb{R}_+$, if for any two initial conditions, $\mathbf{z}^a(0)$, $\mathbf{z}^b(0) \in \mathbb{R}^{N_z}$, and some $\alpha \in \mathbb{R}_+$, the corresponding trajectories $\mathbf{z}^a$ and $\mathbf{z}^b$ generated by the dynamics $f$ satisfy:*

$$\left\| \mathbf{z}^a(t) - \mathbf{z}^b(t) \right\| \leq \alpha \, e^{-\gamma t} \left\| \mathbf{z}^a(0) - \mathbf{z}^b(0) \right\| \quad \forall t \geq 0, \tag{2}$$

*where $\| \cdot \|$ denotes an $L_p$ norm.*

Definition 2.1 implies that a contractive DS guarantees robustness by exponentially *forgetting* perturbations over time. The contraction rate $\gamma$ controls the speed of this process, with larger values indicating faster contraction. We define contractivity using the $L_2$ (Euclidean) norm.

The key question is: How can we learn an expressive and contractive DS with a desired contraction rate? We start by adopting the unconstrained parameterization of continuous-time contracting DSs introduced by Martinelli et al. (2023). In this approach, the dynamics $f$ is modeled as a recurrent equilibrium network (REN), expressed as:

$$\begin{bmatrix} \dot{\mathbf{z}}(t) \\ \boldsymbol{v}(t) \end{bmatrix} = \Omega(\boldsymbol{\theta}, \gamma) \begin{bmatrix} \mathbf{z}(t) \\ \sigma\big(\boldsymbol{v}(t)\big) \end{bmatrix}, \tag{3}$$

where $\boldsymbol{v} \in \mathbb{R}^{N_v}$ is an internal variable initialized at zero, $\gamma$ is the contraction rate, and $\boldsymbol{\theta} \in \mathbb{R}^{N_\theta}$ is the model parameters. The nonlinear function $\sigma : \mathbb{R} \to \mathbb{R}$ is applied element-wise and must be piecewise differentiable with first derivatives restricted to the interval $[0, 1]$. The mapping $\Omega : \mathbb{R}^{N_\theta} \times \mathbb{R}_+ \to \mathbb{R}^{(N_z+N_v) \times (N_z+N_v)}$ ensures that $\boldsymbol{v}(t)$ has a unique value for every $t \geq 0$[1], and that $\mathbf{z}$ satisfies the contractivity criteria in Eq. 2 for every $\boldsymbol{\theta} \in \mathbb{R}^{N_\theta}$. The contraction rate $\gamma$ can be trained or adjusted, as discussed in App. C.1, and influences the resulting model through $\Omega$. Note that RENs can represent arbitrarily deep neural models by selecting the structure of the matrix $\Omega(\boldsymbol{\theta}, \gamma)$, thereby offering a high expressive power (Revay et al., 2023). Further background on contracting RENs and related proofs are provided in App. D.

## 2.3 Expressive Contractive Policies

We formulate the policy $\phi_{\boldsymbol{\theta}}$ using a parameterized autonomous DS in continuous time, given by:

$$\phi_{\boldsymbol{\theta}} : \begin{cases} \hat{\mathbf{y}}(t) = g_{\boldsymbol{\theta}}\big(\mathbf{z}(t)\big) & \textit{(output transformation)} \\ \mathbf{z}(0) = h_{\boldsymbol{\theta}}\big(\mathbf{y}_0\big) & \textit{(initial condition)} \\ \dot{\mathbf{z}}(t) = f_{\boldsymbol{\theta}}\big(\mathbf{z}(t)\big) & \textit{(latent dynamics)} \end{cases}, \tag{4}$$

where $\boldsymbol{\theta} \in \mathbb{R}^{N_\theta}$ collects all parameters.

Given an initial state $\mathbf{y}_0$, the DS $\phi_{\boldsymbol{\theta}}$ evolves through a latent state $\mathbf{z}(t) \in \mathbb{R}^{N_z}$, initialized through $h_{\boldsymbol{\theta}} : \mathbb{R}^{N_y} \to \mathbb{R}^{N_z}$ and governed by the latent dynamics $f_{\boldsymbol{\theta}} : \mathbb{R}^{N_z} \to \mathbb{R}^{N_z}$. The latent state $\mathbf{z}(t)$ is mapped by $g_{\boldsymbol{\theta}} : \mathbb{R}^{N_z} \to \mathbb{R}^{N_y}$ to $\hat{\mathbf{y}}(t) \in \mathbb{R}^{N_y}$, which is the state planned by the policy. A low-level controller then converts this planned state into joint torque or velocity commands for the robot. Together, the DS $\phi_{\boldsymbol{\theta}}$ and the low-level controller form our imitation policy, which maps states to actions. For simplicity, we refer to the DS as the policy. In the sequel, we describe different components of Eq. 4

**Latent dynamics.** We model the latent dynamics with a contractive REN described in Eq. 3. Employing a high-dimensional latent state, $N_z > N_y$, increases the flexibility of our representation, as we empirically show in App. E.3. On the other hand, a low-dimensional latent state, $N_z < N_y$, facilitates learning in some high-dimensional settings (Mohammadi et al., 2024).

**Output transformation.** Owing to the architecture of $f_{\boldsymbol{\theta}}$, all solutions in the latent space converge to a single one, regardless of the initial condition. Yet, it is crucial to maintain the same contraction property for the robot's state trajectories, $\hat{\mathbf{y}}$. In essence, $g_{\boldsymbol{\theta}}$ is required to preserve contractivity, so that $\hat{\mathbf{y}}$ satisfies Eq. 2, for some constant $\alpha$ and $\gamma$. The same behavior is not guaranteed in the original REN formulation (Revay et al., 2023; Martinelli et al., 2023).

To address this, we first change the dimension from the latent space to the state space by applying a learnable linear projection, $\boldsymbol{P}_{\boldsymbol{\theta}} \mathbf{z}$, where $\boldsymbol{P}_{\boldsymbol{\theta}} \in \mathbb{R}^{N_y \times N_z}$. Next, this transformation is followed by $K$ coupling layers, $g_{\boldsymbol{\theta},1} \circ \cdots \circ g_{\boldsymbol{\theta},K}$, which are trainable bijective mappings from $\mathbb{R}^{N_y}$ to $\mathbb{R}^{N_y}$ (Tabak & Turner, 2013; Rana et al., 2020). The resulting output transformation is described by:

$$\hat{\mathbf{y}}(t) = g_{\boldsymbol{\theta}}\big(\mathbf{z}(t)\big) \triangleq g_{\boldsymbol{\theta},1}\big(\cdots \big(g_{\boldsymbol{\theta},K}\big(\boldsymbol{P}_{\boldsymbol{\theta}}\,\mathbf{z}(t)\big)\big)\big), \tag{5}$$

which preserves contractivity according to the next proposition.

**Proposition 2.1.** *If the latent state $\mathbf{z}$ satisfies the contractivity condition in Eq. 2 with $L_2$ norm, then any state trajectory $\hat{\mathbf{y}}$ obtained by the output map in Eq. 5 also satisfies this condition for every parameter $\boldsymbol{\theta} \in \mathbb{R}^{N_\theta}$. Moreover, the contraction rate $\gamma$ is invariant under the output transformation.*

---

[1]While Eq. 3 is not immediately in the form $\dot{\mathbf{z}}(t) = f\big(\mathbf{z}(t)\big)$, it implicitly specifies the relation between $\dot{\mathbf{z}}$ and $\mathbf{z}$ since $\boldsymbol{v}$ is uniquely determined from $\mathbf{z}$.

The proof can be found in App. A.1. This proposition ensures that the mapping $g_{\boldsymbol{\theta}}$ from the latent space to the state space preserves the desired contractivity property.

**Initial condition.** The last element in Eq. 4 to specify is the initial latent state $\mathbf{z}(0)$, which must be set to place the initial state $\hat{\mathbf{y}}(0)$ as close as possible to the desired $\mathbf{y}_0$. Using Eq. 5, this requirement is translated into $\mathbf{y}_0 \approx \hat{\mathbf{y}}(0) = g_{\boldsymbol{\theta}}(\mathbf{z}(0))$. Bearing in mind that the bijective maps are invertible, we can express this relationship as $\boldsymbol{P}_{\boldsymbol{\theta}}\,\mathbf{z}(0) \approx g_{\boldsymbol{\theta},K}^{-1}(\cdots g_{\boldsymbol{\theta},1}^{-1}(\mathbf{y}_0))$. Then, a plausible solution is determined by $\boldsymbol{P}_{\boldsymbol{\theta}}$'s pseudoinverse:

$$\mathbf{z}(0) = h_{\boldsymbol{\theta}}(\mathbf{y}_0) = \boldsymbol{P}_{\boldsymbol{\theta}}^{\dagger}\, g_{\boldsymbol{\theta},K}^{-1}(\cdots g_{\boldsymbol{\theta},1}^{-1}(\mathbf{y}_0)). \tag{6}$$

When $\boldsymbol{P}_{\boldsymbol{\theta}}$ has full column rank, the pseudoinverse coincides with the left inverse, resulting in $\hat{\mathbf{y}}(0) = \mathbf{y}_0$. Otherwise, Eq. 6 provides the least-squares approximation to $\mathbf{y}_0$ (Penrose, 1956).

**Policy architecture.** At this stage, we integrate the essential modules discussed earlier to construct the final policy, by substituting Eq. 3, Eq. 5, and Eq. 6 into the generic policy in Eq. 4:

$$\phi_{\boldsymbol{\theta}} : \begin{cases} \hat{\mathbf{y}}(t) = g_{\boldsymbol{\theta},1}(\cdots(g_{\boldsymbol{\theta},K}(\boldsymbol{P}_{\boldsymbol{\theta}}\,\mathbf{z}(t))) & \text{(output map)} \\ \mathbf{z}(0) = \boldsymbol{P}_{\boldsymbol{\theta}}^{\dagger}\, g_{\boldsymbol{\theta},K}^{-1}(\cdots g_{\boldsymbol{\theta},1}^{-1}(\mathbf{y}_0)) & \text{(initial condition)} \\ \begin{bmatrix} \dot{\mathbf{z}}(t) \\ \boldsymbol{v}(t) \end{bmatrix} = \Omega(\boldsymbol{\theta}, \gamma) \begin{bmatrix} \mathbf{z}(t) \\ \sigma(\boldsymbol{v}(t)) \end{bmatrix} & \text{(latent dynamics)} \end{cases} . \tag{7}$$

The policy is determined by its parameters $\boldsymbol{\theta}$ and the contraction rate $\gamma$. In particular, for any desired $\gamma \in \mathbb{R}_+$, the policy $\phi_{\boldsymbol{\theta}}$ remains contractive regardless of the choice of $\boldsymbol{\theta} \in \mathbb{R}^{N_{\theta}}$. Therefore, $\boldsymbol{\theta}$ can be optimized without imposing any constraints through computationally efficient automatic differentiation tools. In the next section, we learn $\phi_{\boldsymbol{\theta}}$ to effectively imitate the expert and build on the scalability of our method to learn intricate expert behaviors in high-dimensional spaces.

## 3 LEARNING CONTRACTIVE POLICIES THROUGH IMITATION

Each expert trajectory can be intuitively considered as the solution to an unknown initial value problem (IVP). We seek to optimize the policy parameters $\boldsymbol{\theta}$ so that the policy $\phi_{\boldsymbol{\theta}}$ accurately models this IVP. To achieve this, we first explore solving the IVP in a differentiable way, then introduce a loss function to measure the discrepancy between generated and expert trajectories, and finally optimize the policy to minimize this deviation through backpropagation.

### 3.1 DIFFERENTIABLE ODE SOLUTIONS

Consider the IVP in Eq. 7 with an initial condition $\hat{\mathbf{y}}_0 \in \mathcal{Y}$, which we refer to as $ivp(\phi_{\boldsymbol{\theta}},\ \hat{\mathbf{y}}_0)$. The generated solution, denoted by $\hat{\mathbf{y}}$, depends on $\boldsymbol{\theta}$. Therefore, the gradient information for the entire trajectory must be kept along the way to be leveraged by automatic differentiation to optimize $\phi_{\boldsymbol{\theta}}$. Luckily, the Neural ODE framework is capable of solving an IVP while efficiently storing the gradient information through implicit differentiation (Chen et al., 2018).

The Neural ODE framework solves the IVP in Eq. 7 for a selected horizon $H$, which determines the number of integrations. Larger $H$ enables the policy to imitate expert behavior with finer granularity at a higher computational cost (see App. C.2). The IVP solutions, which we synonymously call *policy rollouts*, are formed as differentiable trajectories: $\hat{\mathbf{y}} = (\hat{\mathbf{y}}_0,\ \hat{\mathbf{y}}_1,\ \cdots,\ \hat{\mathbf{y}}_{H-1})$. We further improve the efficiency using batched multiple shooting, which generates an array of rollouts corresponding to different initial conditions in parallel. Next, these policy rollouts are compared to expert demonstrations to train the policy $\phi_{\boldsymbol{\theta}}$.

### 3.2 TRAJECTORY SPACE LOSS

We compare the policy rollout, $\hat{\mathbf{y}}$, with each expert demonstration, $\mathbf{y}^m$, using a discrepancy measure $\ell : \mathcal{Y}^H \times \mathcal{Y}^{N_m} \to \mathbb{R}_{\geq 0}$, where $H$ and $N_m$ are the sizes of $\hat{\mathbf{y}}$ and $\mathbf{y}^m$, respectively. In the simplest case, where the trajectories are of the same size, $H = N_m$ for all $m$, the function $\ell$ can be defined as the mean squared error (MSE). Otherwise, a better alternative is dynamic time warping (DTW) (Keogh & Ratanamahatana, 2005), which can handle trajectories of different lengths.

Indeed, unlike MSE, DTW is not sensitive to time discrepancies: DTW is zero if two trajectories follow the same path but at different speeds. This behavior is desirable because spatial accuracy in robotic tasks is often the primary concern, rather than the execution velocity (Rana et al., 2020; Sochopoulos et al., 2024). We employ the differentiable *soft-DTW* loss (Cuturi & Blondel, 2017) instead of the original formulation, which is tailored for gradient-based optimization (App. C.3).

Next, we introduce the loss function for multiple demonstrations. An intuitive approach is to take a convex combination of the discrepancies associated with each expert trajectory, given by:

$$L(\hat{\mathbf{y}}_0; \boldsymbol{\theta}) \triangleq \sum_{m=1}^{M} \lambda_m(\hat{\mathbf{y}}_0)\, \ell(\hat{\mathbf{y}}, \mathbf{y}^m), \quad \text{s.t. } \hat{\mathbf{y}} = ivp(\phi_{\boldsymbol{\theta}}, \hat{\mathbf{y}}_0), \tag{8}$$

where $\lambda_m(\hat{\mathbf{y}}_0) \in [0, 1]$ is the weight assigned to the difference from the $m$-th demonstration, and these weights satisfy the condition $\sum_{m=1}^{M} \lambda_m(\hat{\mathbf{y}}_0) = 1$. A naive choice for the combination weights is to set them all equal, $\lambda_m(\hat{\mathbf{y}}_0) = \frac{1}{M}$, which results in an average loss. However, this approach can lead to a counterintuitive situation, as illustrated in Fig. 3, where the loss is nonzero even if the policy perfectly replicates one of the expert trajectories, i.e., $L(\mathbf{y}_0^m; \boldsymbol{\theta}) \neq 0$ when $\hat{\mathbf{y}} = \mathbf{y}^m$. Instead, we propose selecting the combination weights inversely proportional to the squared distance between the given initial condition and the initial condition of the $m$-th demonstration,

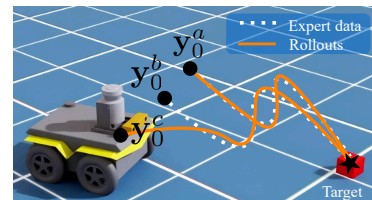

$$\lambda_m(\hat{\mathbf{y}}_0) = \frac{\left(\|\hat{\mathbf{y}}_0 - \mathbf{y}_0^m\|_2^2\right)^{-1}}{\sum_{m'=1}^{M}\left(\|\hat{\mathbf{y}}_0 - \mathbf{y}_0^{m'}\|_2^2\right)^{-1}} \;. \tag{9}$$

The condition $\sum_{m=1}^{M} \lambda_m(\hat{\mathbf{y}}_0) = 1$ can easily be verified for the above weights. In this case, the loss is zero when $\hat{\mathbf{y}} = \mathbf{y}^m$, since only $\lambda_m$ is nonzero, which is the expected behavior.

Figure 3: Uniform weighting averages the deviation from both demonstrations, hence, $L(\mathbf{y}_0^a; \boldsymbol{\theta}) \neq 0$. In contrast, Eq. 9 results in $L(\mathbf{y}_0^a; \boldsymbol{\theta}) = 0$. For $\mathbf{y}_0^c$, the weights in Eq. 9 assign higher importance to the closer $\mathbf{y}_0^b$'s demonstration, which is intuitive.

In real-world scenarios, there is often uncertainty about the initial robot state, $\hat{\mathbf{y}}_0$. A common approach to modeling this uncertainty is to assume that the initial state lies within a set $\mathcal{Y}_0 \subset \mathcal{Y}$ and has the distribution $\rho_0$. Then, the *true* and *empirical* losses, denoted by $\mathcal{L}$ and $\hat{\mathcal{L}}$, are given by:

$$\mathcal{L}(\rho_0; \boldsymbol{\theta}) \triangleq \mathbb{E}_{\hat{\mathbf{y}}_0 \sim \rho_0} L(\hat{\mathbf{y}}_0; \boldsymbol{\theta}) \xrightarrow[\text{estimation}]{\text{unbiased}} \hat{\mathcal{L}}(\mathcal{S}_0; \boldsymbol{\theta}) \triangleq \frac{1}{|\mathcal{S}_0|} \sum_{s=1}^{|\mathcal{S}_0|} L(\hat{\mathbf{y}}_0^s; \boldsymbol{\theta}) \tag{10}$$

where $\mathcal{S}_0 \triangleq \left\{\hat{\mathbf{y}}_0^s\right\}_{s=1}^{|\mathcal{S}_0|}$ is a sampled set of initial states in $\mathcal{Y}_0$. The true loss, $\mathcal{L}$, offers a robust criterion for assessing $\phi_{\boldsymbol{\theta}}$ through averaging out the uncertainty in $\hat{\mathbf{y}}_0$ but is impractical to compute. The empirical loss, $\hat{\mathcal{L}}$, is a computationally efficient approximation of $\mathcal{L}$ and can be minimized to train the policy. In the simple case where $\mathcal{S}_0$ consists of the initial conditions in $\mathcal{D}$, the empirical loss boils down to $\hat{\mathcal{L}}\left(\left\{\mathbf{y}_0^m\right\}_{m=1}^{M}; \boldsymbol{\theta}\right) = \frac{1}{M} \sum_{m=1}^{M} \ell(\hat{\mathbf{y}}^m, \mathbf{y}^m)$, where $\hat{\mathbf{y}}^m$ is the solution to $ivp(\phi_{\boldsymbol{\theta}}, \mathbf{y}_0^m)$. We employ this loss in our experiments.

### 3.3 OPTIMIZATION PROBLEM

The optimal parameters, $\boldsymbol{\theta}^*$, is learned by minimizing the empirical loss $\hat{\mathcal{L}}$ as follows:

$$\boldsymbol{\theta}^* \triangleq \arg\min_{\boldsymbol{\theta} \in \mathbb{R}^{N_\theta}} \hat{\mathcal{L}}\left(\left\{\mathbf{y}_0^m\right\}_{m=1}^{M}; \boldsymbol{\theta}\right), \tag{11}$$

resulting in the optimal policy $\phi_{\boldsymbol{\theta}^*}$. Since the policy parameterization in Subsec. 2.3 is contracting for all parameter choices, we can solve the optimization problem over $\mathbb{R}^{N_\theta}$ without additional constraints. The pseudocode for our method is presented in Alg. 1 and its implementation details are discussed in App. F.

**Remark 3.1.** *The contraction rate $\gamma$ in Alg. 1 can be either set manually or learned as a model parameter. Augmented Lagrangian methods can be used to learn higher contraction rates (App. C.1).*

---

**Algorithm 1** State-only contractive policy learning

---

1: **Require:** Expert data $\mathcal{D}$, contraction rate $\gamma$, learning rate $\eta$, horizon $H$

2: Initialize $\boldsymbol{\theta}$                 ▷ *Policy parameter initialization*
3: **while** not converged **do**

4:   Sample $\hat{\mathbf{y}}_0 \sim \mathbf{y}_0^m \in \mathcal{D}$
5:   Initialize $\mathbf{z}(0) \leftarrow \boldsymbol{P}_{\boldsymbol{\theta}}^{\dagger}\, g_{\boldsymbol{\theta}}^{-1}(\hat{\mathbf{y}}_0)$            ▷ *Latent initial state (Eq. 6)*

6:   $\mathbf{z} := \{\mathbf{z}_t\}_{t=0}^{H-1} \leftarrow \texttt{Neural\_ODE}\big(\phi_{\boldsymbol{\theta}}, \mathbf{z}(0)\big)$    ▷ *Differentiable ODE solver (Subsec. 3.1)*
7:   $\hat{\mathbf{y}} := \{\hat{\mathbf{y}}_t\}_{t=0}^{H-1} \leftarrow g_{\boldsymbol{\theta}}(\mathbf{z})$           ▷ *Output map (Eq. 5)*

8:   **for** each $\mathbf{y}^m$ in $\mathcal{D}$ **do**
9:    Find $\lambda_m$, $\ell(\hat{\mathbf{y}}, \mathbf{y}^m)$            ▷ *Eq. 10, Eq. 9*
10:    Compute $\hat{\mathcal{L}} = \sum_m \lambda_m \ell(\hat{\mathbf{y}}, \mathbf{y}^m)$      ▷ *Empirical loss (Subsec. 3.2)*

11:   Update $\boldsymbol{\theta} \leftarrow \boldsymbol{\theta} - \eta \nabla \hat{\mathcal{L}}$           ▷ *Gradient descent*
12: **return** $\boldsymbol{\theta}$

---

## 4   OUT-OF-SAMPLE PERFORMANCE GUARANTEES

The policy $\phi_{\boldsymbol{\theta}^*}$ is trained to replicate the trajectories in $\mathcal{D}$. Thanks to its contractive behavior, even if the robot starts from an unseen state $\hat{\mathbf{y}}_0 \in \mathcal{Y}_0$ at deployment time, the generated trajectory will converge to those originating within $\mathcal{D}$, likely leading to successful task completion. Still, evaluating the incurred loss $L(\hat{\mathbf{y}}_0;\ \boldsymbol{\theta})$, as defined in Eq. 8, is crucial for ensuring reliability in real-world scenarios. As precisely computing $L$ for every $\hat{\mathbf{y}}_0$ is computationally expensive due to the need to solve $ivp(\phi_{\boldsymbol{\theta}},\ \hat{\mathbf{y}}_0)$, we instead provide an efficient upper bound. To this end, we first introduce an assumption regarding the locality of the initial state.

**Assumption 4.1.** *The initial state $\hat{\mathbf{y}}_0$ lies within a multi-focal ellipse region (Vincze, 1982) with $M$ focal points at the initial conditions in the dataset $\mathcal{D}$, defined as:*

$$\mathcal{Y}_0 = \Big\{ \hat{\mathbf{y}}_0 \Big| \sum_{m=1}^{M} \|\hat{\mathbf{y}}_0 - \mathbf{y}_0^m\|_2 \leq R \Big\}, \tag{12}$$

*where $R \in \mathbb{R}_+$ is a constant scaling the region and $M$ is the number of expert demonstrations.*

Assumption 4.1 requires the initial state to lie within a bounded region relative to those in $\mathcal{D}$, which is a realistic restriction. Under this assumption, we upper-bound the loss for a given initial state.

**Theorem 4.1.** *Consider a contractive policy $\phi_{\boldsymbol{\theta}}$ with contraction rate $\gamma$ and constant $\alpha$ as specified in Definition 2.1. Assume the loss function $\ell$ is the MSE and that all expert demonstrations and policy rollouts have the same length $H$. For $\hat{\mathbf{y}}_0 \in \mathcal{Y}_0$ satisfying Assumption 4.1, it holds that:*

$$L(\hat{\mathbf{y}}_0;\ \boldsymbol{\theta}) \leq \underbrace{\sum_{m=1}^{M} \lambda_m(\hat{\mathbf{y}}_0)\, \ell(\hat{\mathbf{y}}^m, \mathbf{y}^m)}_{(i)} + \underbrace{\frac{\alpha^2\, R^2\, (e^{-2\gamma} - 1)}{H\, M\, (e^{\frac{-2\gamma}{H}} - 1)}}_{(ii)},$$

*where $\mathbf{y}^m$ is the $m$-th demonstration, $\hat{\mathbf{y}}^m$ is the solution to $ivp(\phi_{\boldsymbol{\theta}}, \mathbf{y}_0^m)$, and $L$ is defined in Eq. 8.*

The proof is outlined in App. A.2. Theorem 4.1 validates the intuition that a contractive policy performs effectively in deployment through upper-bounding the incurred loss. The bound consists of two terms. The first term (i) is a weighted sum of MSE between rollouts starting from the initial states in $\mathcal{D}$ and their corresponding demonstrations. These MSE values are precomputed, leaving only the coefficients $\lambda_m[\hat{\mathbf{y}}_0]$ to be calculated for each $\hat{\mathbf{y}}_0$. The second term (ii) accounts for uncertainty in $\hat{\mathbf{y}}_0$. It decreases when $R$ is smaller, reflecting less uncertainty, and when $\phi_{\boldsymbol{\theta}}$ is more contractive—either through a smaller $\alpha$ or a higher $\gamma$—as the influence of $\hat{\mathbf{y}}_0$ fades more quickly. Finally, we present an upper bound on the true loss.

**Corollary 4.1.1.** *Under the same conditions as Theorem 4.1, the true loss $\mathcal{L}$ defined in Eq. 10 is upper-bounded by:*

$$\mathcal{L}(\rho_0;\ \boldsymbol{\theta}) \leq \max_{m \in \{1, \cdots, M\}} \ell(\hat{\mathbf{y}}^m, \mathbf{y}^m) + \frac{\alpha^2\, R^2\, (e^{-2\gamma} - 1)}{H\, M\, (e^{\frac{-2\gamma}{H}} - 1)},$$

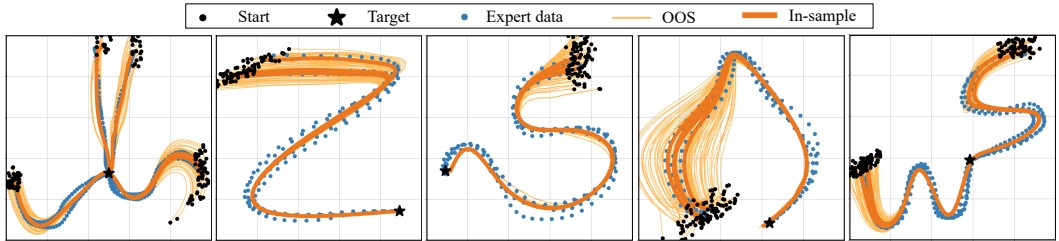

Figure 4: In-sample and OOS policy rollouts for selected 2D tasks in the LASA dataset. The training process promotes higher contraction rates ( App. E.2), resulting in effective OOS recovery.

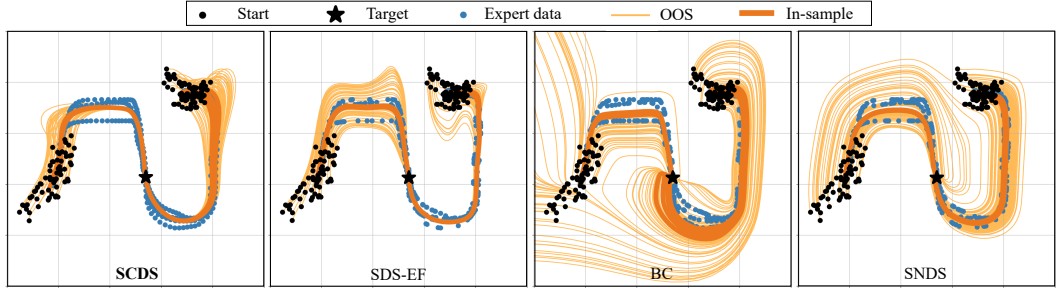

Figure 5: Comparing SCDS to the selected baselines on in-sample and OOS rollouts in the 2D task space. BC results in diverging trajectories due to the lack of stability guarantees. While SNDS and SDS-EF ensure global stability and reach the target, they display large deviations from expert data. In contrast, SCDS efficiently recovers from OOS states through its contracting transient behavior.

*where $\rho_0$ is any distribution over the set $\mathcal{Y}_0$ satisfying Assumption 4.1.*

The proof is provided in App. A.3. Similar to Theorem 4.1, the bound tightens when the policy replicates the demonstrations more closely and the uncertainty level is lower, either due to a smaller $R$ and $\alpha$ or a larger $\gamma$.

## 5 EXPERIMENTS

We conduct empirical studies on the LASA dataset (Khansari-Zadeh & Billard, 2011), as well as the more complex Robomimic dataset (Mandlekar et al., 2021). These datasets are detailed in App. G. We deploy our policies in the Isaac Lab (Orbit) simulator (Mittal et al., 2023) for the Franka Panda manipulator arm and the Clearpath Jackal wheeled robot. A highlight of the results is discussed here, while further experiments, computation time comparisons, and ablation studies on the impact of the contraction rate, the latent space, and the coupling layers are provided in App. E.

### 5.1 SETUP AND BASELINES

**Baselines.** We compare the accuracy of our approach to the following baselines: Stable Neural Dynamical System (SNDS) (Abyaneh et al., 2024), Stable Dynamical System Learning using Euclideanizing Flows (SDS-EF) (Rana et al., 2020), and Behavioral Cloning (BC) (Pomerleau, 1988; Mandlekar et al., 2020). Although BC does not guarantee stability, we include it in our empirical evaluation to provide further ground for comparison. Other baselines are trained using both states and their time derivatives, while SCDS relies solely on state measurements for training. The implementation details and hyperparameters are discussed in App. F.2.

**Evaluation.** After training each policy, we generate two sets of rollouts: one with the initial state drawn from the dataset $\mathcal{D}$ (in-sample rollouts), and another with the initial state not in $\mathcal{D}$ (OOS

Figure 6: In-sample and OOS policy rollouts for the 6D *Can* and *Lift* tasks in the Robomimic dataset, using both 1 and 50 demonstrations. SCDS generates contractive trajectories even in higher-dimensional spaces, where expert trajectories exhibit non-converging patterns.

rollouts). To generate an OOS initial state, we first sample one of the demonstrations uniformly at random and then sample uniformly from a hyper-sphere around it. The hyper-sphere around $\mathbf{y}_0^m$ has a radius of $0.1\|\mathbf{y}_0^m\|_2$.[2] The loss from in-sample rollouts corresponds to the empirical loss defined in Subsec. 3.2, while the OOS loss approximates the true loss given by Eq. 10. A low in-sample loss indicates the policy is expressive, whereas a low OOS loss signifies it generalizes well. We use MSE and soft-DTW as the metrics $\ell$ to capture the difference between rollouts and expert trajectories.

## 5.2 RESULTS AND DISCUSSION

**LASA dataset.** Fig. 4 illustrates OOS rollouts for different expert handwriting demonstrations in the LASA-2D (position) and the LASA-4D (position and velocity) datasets. For all random initial states, the induced trajectories reliably contract towards the expert demonstrations and accurately imitate them with high precision. Next, in Fig. 5, we compare the performance of SCDS with the introduced baselines for one of the motions. As shown, some trajectories generated by BC diverge from the target, which is expected since this baseline lacks stability guarantees. While the stable SDS-EF and SNDS baselines ensure convergence to the target, they lack efficient OOS recovery and deviate from expert data. Unlike the stable baselines, SCDS generates smoothly contracting trajectories that converge to the target, offering a more precise replication of expert demonstrations.

Subsequently, we extend our comparison to all motions in the LASA dataset and summarize the results for in-sample and OOS rollouts in Tab. 1. We also consider a 4-dimensional version of the LASA dataset by replicating both the expert's position and velocity. As shown in the table, SCDS outperforms or produces highly competitive results across all scenarios.

**Robomimic dataset.** We utilize the Robomimic dataset, specifically 6D (task space position and orientation) and 14D (joint position and velocity) expert demonstrations, to highlight the applicability of our method in high-dimensional spaces. Expert trajectories do not necessarily end at the same target, in contrast to the LASA dataset. Fig. 6 shows OOS rollouts for policies trained on single and multiple expert trajectories. When multiple trajectories are used, the contractive policy learns to contract to an average behavior to accomplish the task. Tab. 1 presents results averaged across four tasks: Lift, Can, Square, and Transport. As shown, previous methods tend to overfit the dataset, resulting in high errors on OOS data. In contrast, SCDS demonstrates strong generalization, reducing the best baseline result by up to a factor of 2.5.

**Upper bound.** We compute the upper bound on the true loss when using the MSE metric following Corollary 4.1.1 and report it in Tab. 1. The constant $\alpha$ that appears in the bound is approximated empirically using a Monte Carlo approach. As expected, the MSE upper bound is consistently larger than the observed loss in Tab. 1, yet it remains relatively tight.

**Robot deployment.** Finally, we deploy the trained policies in two sample cases: i) the Robomimic-14D Lift manipulation task using the Franka Panda arm, and ii) a navigation task with the Clearpath Jackal wheeled robot following the Sine motion in the LASA-4D dataset. In both cases, the rollouts are tested in the Isaac Lab (Orbit) simulator (Mittal et al., 2023). The average OOS soft-DTW errors

---

[2]Since this is a bounded region, an encapsulating multi-focal ellipse exists for it.

Table 1: Evaluating in-sample and out-of-sample rollouts error on the LASA and the Robomimic datasets. The lowest (best) values for each metric and dataset are highlighted. The expressive power of SCDS enables it to achieve near state-of-the-art performance on in-sample data. More notably, it consistently outperforms other baselines in OOS recovery, and conforms with the theoretical upper bound for MSE loss, $\mathcal{L}_{ub}^{MSE}$, as derived in Sec. 4.

| Expert | LASA-2D | | LASA-4D | | Robomimic-6D | | Robomimic-14D | |
|---|---|---|---|---|---|---|---|---|
| Metric | MSE | soft-DTW | MSE | soft-DTW | MSE | soft-DTW | MSE | soft-DTW |
| SNDS | $0.02 \pm 0.01$ | $0.72 \pm 0.14$ | $0.03 \pm 0.01$ | $1.04 \pm 0.19$ | $0.65 \pm 0.55$ | $1.26 \pm 0.70$ | $2.42 \pm 1.37$ | $4.15 \pm 0.92$ |
| BC | $0.04 \pm 0.02$ | $0.98 \pm 0.12$ | $0.05 \pm 0.03$ | $1.48 \pm 0.16$ | $0.56 \pm 0.32$ | $1.88 \pm 0.20$ | $1.75 \pm 0.22$ | $4.86 \pm 0.58$ |
| SDS-EF | $0.03 \pm 0.01$ | $0.85 \pm 0.13$ | $0.05 \pm 0.02$ | $1.10 \pm 0.15$ | $0.50 \pm 0.28$ | $1.01 \pm 0.66$ | $3.30 \pm 0.75$ | $5.65 \pm 0.58$ |
| **SCDS** | $0.02 \pm 0.01$ | $0.65 \pm 0.05$ | $0.03 \pm 0.01$ | $0.72 \pm 0.12$ | $0.56 \pm 0.22$ | $1.05 \pm 0.37$ | $1.68 \pm 0.45$ | $4.10 \pm 0.40$ |
| SNDS | $2.73 \pm 1.67$ | $6.91 \pm 1.46$ | $3.65 \pm 2.12$ | $9.85 \pm 0.63$ | $1.58 \pm 0.93$ | $3.27 \pm 1.88$ | $6.88 \pm 2.16$ | $12.17 \pm 2.70$ |
| BC | $8.63 \pm 4.05$ | $16.25 \pm 5.27$ | $19.25 \pm 7.34$ | $27.48 \pm 6.83$ | $11.05 \pm 7.41$ | $19.94 \pm 9.57$ | $44.98 \pm 15.11$ | $37.82 \pm 14.13$ |
| SDS-EF | $1.78 \pm 0.34$ | $7.13 \pm 1.51$ | $2.33 \pm 0.47$ | $10.21 \pm 1.98$ | $1.15 \pm 0.81$ | $2.67 \pm 1.40$ | $11.10 \pm 2.10$ | $10.44 \pm 1.66$ |
| **SCDS** | $0.32 \pm 0.15$ | $1.72 \pm 0.54$ | $1.09 \pm 0.21$ | $2.58 \pm 0.30$ | $0.89 \pm 0.26$ | $1.71 \pm 0.53$ | $2.68 \pm 0.65$ | $6.27 \pm 0.48$ |
| $\mathcal{L}_{ub}^{MSE}$ | $0.49 \pm 0.07$ | | $1.33 \pm 0.14$ | | $1.43 \pm 0.22$ | | $2.91 \pm 0.45$ | |

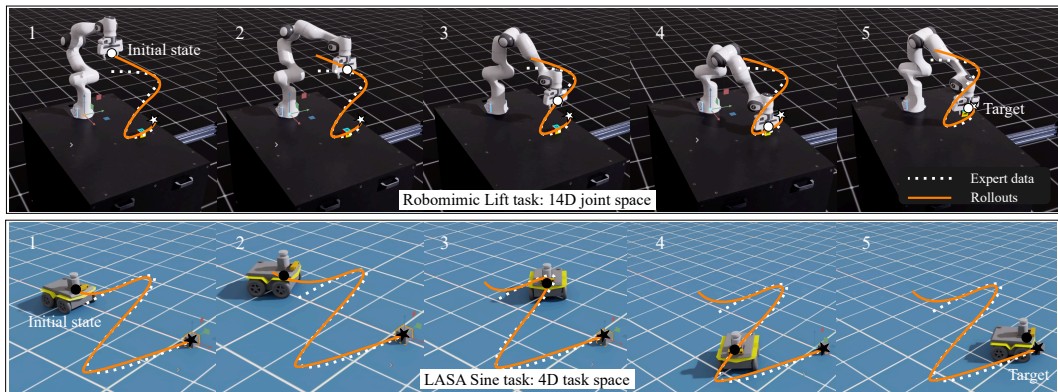

Figure 7: Simulation rollouts with Franka arm and Jackal mobile robots in Isaac Lab. Both robots start from a random OOS initial state and swiftly converge to expert data.

measured over 50 different initial states are $3.68 \pm 0.33$ and $1.94 \pm 0.37$, respectively. Snapshots of random rollouts are illustrated in Fig. 7.

## 6 CONCLUSION, LIMITATIONS, AND FUTURE WORK

We introduced SCDS: a neural contractive imitation policy that ensures all trajectories contract over time and ultimately converge to the target. SCDS offers robustness to OOS states throughout the transient phase, yielding stronger reliability than asymptotic stability. By parameterizing only contractive policies, SCDS learns the policy through efficient unconstrained optimization. Performance is further enhanced by learning in a high-dimensional latent space and employing coupling layers. Finally, we provide generalization guarantees when the robot starts from unseen initial conditions.

**Limitations.** Learning from rapidly changing expert data demands a longer horizon, and thus more computation. As such, there exists a trade-off between computational complexity and imitation accuracy for intricate expert behavior (App. C.2).

**Future work.** Subsequent research may extend the current method to long-horizon expert trajectories while maintaining a reasonable computational load and the contractive nature of the policy. Additionally, RENs can be replaced by other parameterized contraction models. The applicability of SCDS to other robotic tasks, such as quadrupeds and drones, is straightforward, and SCDS can imitate intricate expert behavior in these scenarios. Finally, applications of contraction theory to stochastic imitation policies (Fu et al., 2018) can be explored.

ACKNOWLEDGMENTS

This work was supported as a part of NCCR Automation, a National Centre of Competence in Research, funded by the Swiss National Science Foundation (grant number 51NF40_225155) and the NECON project (grant number 200021219431).

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

# Appendix

## Table of Contents

## A MATHEMATICAL PROOFS

### A.1 PROPOSITION 2.1

We first state and prove a classic linear algebra lemma.

**Lemma A.1.** *Let $\boldsymbol{P} \in \mathbb{R}^{m \times n}$ be an arbitrary matrix and $\sigma_{min}(\boldsymbol{P})$ denote its smallest singular value. For every vector $\boldsymbol{v} \in \mathbb{R}^n$, it holds that:*

$$\sigma_{min}(\boldsymbol{P})^2 \leq \frac{\|\boldsymbol{P}\boldsymbol{v}\|_2^2}{\|\boldsymbol{v}\|_2^2}.$$

*Proof.* We start by expanding the right-hand side, as $\|\boldsymbol{P}\boldsymbol{v}\|_2^2 / \|\boldsymbol{v}\|_2^2 = \boldsymbol{v}^T \boldsymbol{P}^T \boldsymbol{P} \boldsymbol{v} / \|\boldsymbol{v}\|_2^2$. By the Courant–Fischer Theorem[3], we obtain $\lambda_{min}(\boldsymbol{P}^T \boldsymbol{P}) \leq \boldsymbol{v}^T \boldsymbol{P}^T \boldsymbol{P} \boldsymbol{v} / \|\boldsymbol{v}\|_2^2$. Finally, since the singular values of $\boldsymbol{P}$ are the square roots of the eigenvalues of $\boldsymbol{P}^T \boldsymbol{P}$, one has $\lambda_{min}(\boldsymbol{P}^T \boldsymbol{P}) = \sigma_{min}(\boldsymbol{P})^2$. This completes the proof. $\square$

Below, we present the proof of Proposition 2.1.

*Proof.* Let $\hat{\mathbf{y}}_0^a$ and $\hat{\mathbf{y}}_0^b \in \mathbb{R}^{N_y}$ represent two arbitrary initial states, and denote the state and latent state trajectories obtained by solving Eq. 7 as $\mathbf{y}^a, \mathbf{y}^b$ and $\mathbf{z}^a, \mathbf{z}^b$, respectively. The proof proceeds in two steps. First, we show that the linear projection preserves the contraction behavior at the same rate. We arrive at the following inequality through the following algebraic operations:

$$
\begin{aligned}
\left\|\boldsymbol{P_\theta}\, \mathbf{z}^a(t) - \boldsymbol{P_\theta}\, \mathbf{z}^b(t)\right\| &\leq \left\|\boldsymbol{P_\theta}\right\| \left\|\mathbf{z}^a(t) - \mathbf{z}^b(t)\right\| && \textit{(norm definition)} \\
&\leq \alpha \left\|\boldsymbol{P_\theta}\right\| e^{-\gamma t} \left\|\mathbf{z}^a(0) - \mathbf{z}^b(0)\right\| && \textit{(\textbf{z} satisfies Eq. 2)} \\
&\leq \frac{\alpha \left\|\boldsymbol{P_\theta}\right\|}{\sigma_{min}^2(\boldsymbol{P_\theta})} e^{-\gamma t} \left\|\boldsymbol{P_\theta}\, \mathbf{z}^a(0) - \boldsymbol{P_\theta}\, \mathbf{z}^b(0)\right\|, && \textit{(Lemma A.1)} \quad (13)
\end{aligned}
$$

where $\sigma_{min}(\boldsymbol{P_\theta})$ denotes the smallest singular value of $\boldsymbol{P_\theta}$. Hence, with $\alpha' = \frac{\alpha \|\boldsymbol{P_\theta}\|}{\sigma_{min}^2(\boldsymbol{P_\theta})}$ as the new constant, it follows from Eq. 13 that $\boldsymbol{P_\theta}\, \mathbf{z}$ satisfies the condition in Eq. 2.

Furthermore, it is established in (Manchester & Slotine, 2015) that bijective maps preserve contractivity. It can also be concluded from the proof in (Manchester & Slotine, 2015) that $\gamma$ *remains unchanged*. Combined with the first step, this proves the proposition. $\square$

### A.2 THEOREM 4.1

We first state and prove a basic inequality.

**Lemma A.2.** *For every $x_1, \cdots, x_M \in \mathbb{R}_+$, it holds that $\frac{M}{\sum_{m=1}^{M} 1/x_m} \leq \frac{\sum_{m=1}^{M} x_m}{M}$.*

*Proof of Lemma A.2.* We begin by rewriting the lemma as follows:

$$
\frac{M}{\sum_{m=1}^{M} \frac{1}{x_m}} \leq \frac{\sum_{m=1}^{M} x_m}{M} \iff 1 \leq \frac{1}{M^2} \sum_{m=1}^{M} \left(\frac{1}{x_m}\right) \left(\sum_{m=1}^{M} x_m\right)
$$

$$
\iff 1 \leq \frac{1}{M^2} \sum_{m=1}^{M} \sum_{m'=1}^{M} \frac{x_{m'}}{x_m}. \quad (14)
$$

Since $x_m \in \mathbb{R}_+$ for all $m \in \{1, \cdots, M\}$, we apply the inequality of arithmetic and geometric means[4] to derive:

$$
\frac{1}{M^2} \sum_{m=1}^{M} \sum_{m'=1}^{M} \frac{x_{m'}}{x_m} \geq \left(\prod_{m=1}^{M} \prod_{m'=1}^{M} \frac{x_{m'}}{x_m}\right)^{1/M^2} = 1. \quad (15)
$$

---

[3]For every square matrix $\boldsymbol{A} \in \mathbb{R}^{n \times n}$ with smallest eigenvalue $\lambda_{min}(\boldsymbol{A})$ and every vector $\boldsymbol{v} \in \mathbb{R}^n$, it holds that $\lambda_{min}(\boldsymbol{A}) \leq \boldsymbol{v}^T \boldsymbol{A} \boldsymbol{v} / \|\boldsymbol{v}\|_2^2$.

[4]This inequality states that the arithmetic mean of a list of non-negative real numbers is greater than or equal to the geometric mean of the same list.

This inequality satisfies (14), thereby completing the proof. $\qquad\square$

*Proof of Theorem 4.1.* Using Eq. 8 and the triangle inequality for MSE, we have:

$$
\begin{aligned}
L(\hat{\mathbf{y}}_0;\,\boldsymbol{\theta}) &= \sum_{m=1}^{M} \lambda_m(\hat{\mathbf{y}}_0)\,\ell\big(\hat{\mathbf{y}},\,\mathbf{y}^m\big) \\
&\leq \sum_{m=1}^{M} \lambda_m(\hat{\mathbf{y}}_0)\,\ell\big(\hat{\mathbf{y}},\,\hat{\mathbf{y}}^m\big) + \sum_{m=1}^{M} \lambda_m(\hat{\mathbf{y}}_0)\,\ell(\hat{\mathbf{y}}^m,\mathbf{y}^m),
\end{aligned}
\tag{16}
$$

where $\hat{\mathbf{y}}$ is the solution to $ivp(\phi_{\boldsymbol{\theta}},\hat{\mathbf{y}}_0)$ for an arbitrary $\hat{\mathbf{y}}_0 \in \mathcal{Y}_0$. We expand the first loss term, assuming MSE loss is used:

$$
\begin{aligned}
\ell(\hat{\mathbf{y}},\,\hat{\mathbf{y}}^m) &= \frac{1}{H}\sum_{i=0}^{H-1}\big\|\hat{\mathbf{y}}_i - \hat{\mathbf{y}}_i^m\big\|_2^2 \\
&\leq \frac{1}{H}\sum_{i=0}^{H-1}\alpha^2 e^{-2\gamma i/H}\big\|\hat{\mathbf{y}}_0 - \hat{\mathbf{y}}_0^m\big\|_2^2 \\
&= \frac{\alpha^2\left(e^{-2\gamma}-1\right)}{H\left(e^{-2\gamma/H}-1\right)}\big\|\hat{\mathbf{y}}_0 - \hat{\mathbf{y}}_0^m\big\|_2^2,
\end{aligned}
\tag{17}
$$

where the inequality follows from the contractivity definition in Eq. 2 evaluated at time $t = i/H$. We then simplify the first term in Eq. 16 using Eq. 17 and the formula for $\lambda_m$ given by Eq. 9, keeping in mind that $\hat{\mathbf{y}}_0^m = \mathbf{y}_0^m$:

$$
\sum_{m=1}^{M} \lambda_m(\hat{\mathbf{y}}_0)\,\ell(\hat{\mathbf{y}},\,\hat{\mathbf{y}}^m) \leq \frac{\alpha^2\left(e^{-2\gamma}-1\right)}{H\left(e^{-2\gamma/H}-1\right)} \cdot \frac{M}{\sum_{m=1}^{M}\left(\|\hat{\mathbf{y}}_0 - \mathbf{y}_0^m\|_2^2\right)^{-1}}\ .
\tag{18}
$$

Next, by applying Lemma A.2, we obtain:

$$
\frac{M}{\sum_{m=1}^{M}\frac{1}{\|\hat{\mathbf{y}}_0 - \mathbf{y}_0^m\|_2^2}} \leq \frac{\sum_{m=1}^{M}\|\hat{\mathbf{y}}_0 - \mathbf{y}_0^m\|_2^2}{M} \leq \frac{\left(\sum_{m=1}^{M}\|\hat{\mathbf{y}}_0 - \mathbf{y}_0^m\|_2\right)^2}{M} \leq \frac{R^2}{M},
\tag{19}
$$

where the last inequality follows from the conic initial set, $\hat{\mathbf{y}}_0 \in \mathcal{Y}_0$, in Assumption 4.1. Combining Eq. 16, Eq. 18, and Eq. 19 completes the proof:

$$
L(\hat{\mathbf{y}}_0;\,\boldsymbol{\theta}) \leq \frac{\alpha^2\left(e^{-2\gamma}-1\right)}{H\left(e^{-2\gamma/H}-1\right)} \cdot \frac{R^2}{M} + \sum_{m=1}^{M} \lambda_m(\hat{\mathbf{y}}_0)\,\ell(\hat{\mathbf{y}}^m,\mathbf{y}^m).
\tag{20}
$$

$\qquad\square$

## A.3 COROLLARY 4.1.1

*Proof.* First, we use the fact that the convex combination of some values is smaller than their maximum to obtain from Theorem 4.1 that:

$$
L(\hat{\mathbf{y}}_0;\,\boldsymbol{\theta}) \leq \max_{m\in\{1,\cdots,M\}} \ell(\hat{\mathbf{y}}^m,\mathbf{y}^m) + \frac{\alpha^2 R^2\left(e^{-2\gamma}-1\right)}{H M\left(e^{-2\gamma/H}-1\right)},
$$

for all $\hat{\mathbf{y}}_0 \in \mathcal{Y}_0$. Next, notice that if a bound holds for every $\hat{\mathbf{y}}[0] \in \mathcal{Y}_0$, it also holds when taking the expectation with respect to any distribution over $\hat{\mathbf{y}}[0]$, i.e.:

$$
\mathcal{L}(\rho_0;\,\boldsymbol{\theta}) = \mathop{\mathbb{E}}_{\hat{\mathbf{y}}_0 \sim \rho_0} L(\hat{\mathbf{y}}_0;\,\boldsymbol{\theta}) \leq \max_{m\in\{1,\cdots,M\}} \ell(\hat{\mathbf{y}}^m,\mathbf{y}^m) + \frac{\alpha^2 R^2\left(e^{-2\gamma}-1\right)}{H M\left(e^{-2\gamma/H}-1\right)}.
\tag{21}
$$

Since the left-hand side is the definition of the true loss, $\mathcal{L}(\rho_0;\,\boldsymbol{\theta})$, the proof is completed. $\qquad\square$

## B    EXTENDED RELATED WORK

We introduced contractive imitation policies in Sec. 1, and an overview of these methods is provided in Tab. 2. In this section, we extend the discussion to imitation policies in a broader context. Specifically, we review both stable and unstable imitation policies.

Table 2: Comparing contractive DS methods for imitation learning. Except for NCDS and SCDS, the rest attempt to solve a computationally challenging constrained optimization problem.

| | Unconstrained optimization | Adjustable contraction rate | State-only | Learn in latent space |
|---|---|---|---|---|
| C-GMR (Blocher et al., 2017) | X | X | X | X |
| CDSP (Ravichandar et al., 2017) | X | X | X | X |
| CVF (Sindhwani et al., 2018) | X | X | X | X |
| NCDS (Mohammadi et al., 2024) | ✓ | X | X | only lower-dimensional |
| **SCDS (ours)** | ✓ | ✓ | ✓ | ✓ |

### B.1    STABLE DYNAMICAL POLICIES WITH RESTRICTED REPRESENTATIONS

A common approach to ensuring global stability is through Lyapunov theory (Devaney, 2021). In this framework, a Lyapunov candidate function must be defined at each step of policy optimization, ensuring that the policy satisfies Lyapunov conditions with respect to this candidate. To increase the flexibility of these methods, the Lyapunov candidate is typically learned rather than fixed a priori.

However, jointly learning both the policy and the Lyapunov candidate is computationally intensive. To alleviate this, prior efforts have often limited the hypothesis class of the policy or the Lyapunov candidate, which in turn reduces the expressiveness and accuracy of imitation. In this context, Khansari-Zadeh & Billard (2011) employs a fixed quadratic Lyapunov candidate with a dynamical policy represented by a Gaussian mixture model. Nevertheless, trajectories with non-decreasing distance from the equilibrium are not consistent with this Lyapunov function and hence, cannot be achieved.

To address this limitation, Khansari-Zadeh & Billard (2014) and Abyaneh & Lin (2023) introduce trainable parameterized Lyapunov candidates, while Neumann & Steil (2015) uses diffeomorphisms to achieve similar goals. Despite these improvements, the expressive power remains limited. Additionally, scalability to higher-dimensional spaces remains a challenge, primarily due to the complexity of the underlying non-convex optimization.

### B.2    STABLE NEURAL DYNAMICAL POLICIES

Recent advancements in globally stable imitation learning have explored using neural networks to model policies, enhancing their expressive power. To ensure stability, a line of research has focused on learning a simple stable DS and mapping it to a more complex one through a trainable diffeomorphism (Rana et al., 2020; Zhang et al., 2022). A diffeomorphism is an invertible map that preserves stability and enables the generation of highly nonlinear policies. While these techniques have shown potential even for complex motion tasks, they have several limitations: they require a large number of demonstrations during training, tend to produce policies that are quasi-stable, and, most importantly, struggle to learn in high-dimensional state spaces due to the need to preserve the invertibility of the diffeomorphic transformation.

Another class of methods (Kolter & Manek, 2019; Sochopoulos et al., 2024; Abyaneh et al., 2024) employs projection at each training step to enforce stability constraints. These projection-based approaches have the advantage of requiring fewer demonstrations. However, projection does not inherently guarantee the smoothness of the resulting policy. As a result, these policies can be difficult to train, particularly in high-dimensional spaces with rapidly changing dynamics.

### B.3    IMITATION POLICIES WITH NO STABILITY GUARANTEES

A parallel research direction focuses on learning highly expressive policies without stability guarantees. In this context, recent advancements in apprenticeship learning (Abbeel & Ng, 2004) and

inverse reinforcement learning (Ziebart et al., 2008) can be leveraged for learning imitation policies. For instance, GAIL (Ho & Ermon, 2016), AIRL (Fu et al., 2018), and VAIL (Peng et al., 2019) introduce an adversarial IRL framework, with AIRL focusing on robustness to dynamic changes and VAIL improving training stability. DART (Laskey et al., 2017) adds noise to the training data to mitigate compounding errors in "off-policy" methods. Dagger-style approaches (Ross & Bagnell, 2010; Menda et al., 2019) iteratively aggregate new data from both expert and novice policies, offering some level of safety assurance. However, the absence of stability guarantees in these methods means they offer no out-of-sample guarantees, which limits their applicability in real-world robotic environments

### B.4 RECURRENT EQUILIBRIUM NETWORKS (RENS)

In this work, we employed RENs to model the dynamics of our policy. RENs were introduced in the discrete-time setup by Revay et al. (2023) as a broad class of nonlinear dynamical models with built-in stability and reliability guarantees. Subsequently, Martinelli et al. (2023) extended this idea to continuous-time models, proving similar characteristics. In addition to being contraction by design, RENs are highly flexible. For instance, they can represent all stable linear systems, all previously known sets of contracting recurrent neural networks, and all deep feedforward neural networks (Revay et al., 2023). RENs have been employed in various applications, such as system identification (Revay et al., 2023; Martinelli et al., 2023), control systems (Boroujeni et al., 2024), and optimization (Martin & Furieri, 2024).

## C  Design considerations

SCDS involves design choices that balance accuracy with computational efficiency. In this section, we highlight the key design parameters that impact this trade-off and offer guidance on how to select them for achieving optimal performance.

### C.1  Contraction rate

As highlighted in Sec. 2, the contraction rate $\gamma$ can either be fixed or learned alongside the model parameters. Higher values of $\gamma$ enforce more restrictive behavior, causing trajectories to contract exponentially faster, according to the definition of contraction theory Definition 2.1. This introduces the first key trade-off: while higher contraction rates lead to faster out-of-sample recovery and improved robustness to unseen data, they also reduce the model's expressiveness.

The bijection block in the output map, presented in Eq. 5, helps mitigate this reduction in expressivity. In our experiments, we achieve high contraction rates with bijection blocks of reasonable sizes, maintaining a balance between contraction and policy expressivity. The specific hyperparameters used for the LASA and Robomimic datasets are detailed in Tab. 7.

### C.2  Horizon

Extending the horizon length, denoted as $H$, necessitates additional integrations by the ODE solver, which increases the computational complexity of the method. This is exemplified by a differentiable Euler integration step, expressed as:

$$\mathbf{z}_{t+1} = \mathbf{z}_t + f_{\boldsymbol{\theta}}(\mathbf{z}_t)\Delta t, \quad \forall t \in 0, 1, \ldots, H. \tag{22}$$

The need to retain gradient information across all time steps further adds to memory and computational demands. However, extending the horizon may be imperative to capture long-horizon or fast-evolving expert behaviors. In cases where long horizons are crucial—though not required for the Robomimic and LASA datasets—using more advanced integrators and reducing solver tolerance can help manage computational costs. The horizon lengths for the LASA and the Robomimic datasets are given in Tab. 7.

### C.3  Soft DTW

The computation of Dynamic Time Warping (DTW) using its original formulation (Keogh & Ratanamahatana, 2005) is known to be resource-intensive. To address this, soft-DTW (Cuturi & Blondel, 2017) presents a more computationally efficient alternative while still yielding acceptable precision. The soft-DTW algorithm assumes a matrix of admissible paths $\mathcal{A}$, and defines the loss function as follows:

$$\text{soft-}DTW^{\beta}(\hat{\mathbf{y}}, \mathbf{y}^m) = \min_{\tau \in \mathcal{A}(\hat{\mathbf{y}}, \mathbf{y}^m)}{}^{\beta} \sum_{(i,j) \in \tau} d(\hat{y}_i, y_j^m)^2, \tag{23}$$

where $\min^{\beta}$ denotes the soft-min operator with a smoothing factor $\beta$, and $d$ represents a distance metric, such as the Euclidean distance. The soft mean operator is defined by $\min^{\beta}(a_1, \ldots, a_n) = -\beta \log \sum_i e^{-a_i/\beta}$. As $\beta \to 0^+$, the soft-DTW loss converges to the original DTW loss. Importantly, unlike standard DTW, the soft-DTW variant is *differentiable* across its entire domain, making it suitable for use as a loss function in model training via backpropagation.

### C.4  Implicit depth and invertible layers

The expressiveness of the REN architecture can be increased by increasing the dimension of $\boldsymbol{v}$ in Eq. 3, which deepens the implicit layer (Martinelli et al., 2023). However, our empirical observations revealed that this approach substantially increases training time. Consequently, we chose to keep this parameter fixed and instead enhance representation power through the bijection block. Particularly, we show that increasing the number of invertible (bijective) layers enhances the representation power in App. E.4. The efficient implementation of these layers, covered in App. F.1, reduces computational overhead, making the additional computations feasible.

## D  CONTINUOUS REN PROPERTIES

We provide the corresponding theory, stating that continuous RENs are contractive for *any* choice of parameters. We encourage referring to the original paper (Martinelli et al., 2023; Revay et al., 2023) to fully understand the mechanism behind the REN's contractive structure. However, highlights of the core theorems are briefly glossed over in this section.

### D.1  DETAILED STRUCTURE

We show an abstract format of the original REN formulation in Eq. 3. The exact formulation is given by,

$$
\begin{bmatrix} \dot{x}(t) \\ v(t) \\ y(t) \end{bmatrix} = \underbrace{\begin{bmatrix} A & B_1 & B_2 \\ C_1 & D_{11} & D_{12} \\ C_2 & D_{21} & D_{22} \end{bmatrix}}_{\tilde{A}} \begin{bmatrix} x(t) \\ w(t) \\ u(t) \end{bmatrix} + \underbrace{\begin{bmatrix} b_x \\ b_v \\ b_y \end{bmatrix}}_{\tilde{b}}, \tag{24}
$$

$$
w(t) = \sigma(v(t)), \tag{25}
$$

where $x(t) \in \mathbb{R}^n$, $u(t) \in \mathbb{R}^m$ and $y(t) \in \mathbb{R}^p$ are respectively the state ($\mathbf{z}$ in our paper), the input, and output ($\mathbf{y}$ in our paper) at time $t$. The function $\sigma(\cdot)$ is an entry-wise nonlinear activation function. The input and output of $\sigma(\cdot)$ are $v(t)$, $w(t) \in \mathbb{R}^q$, respectively.

First, we restate the Eq. 3 used in the main text to describe RENs:

$$
\begin{bmatrix} \dot{\mathbf{z}}(t) \\ \boldsymbol{v}(t) \end{bmatrix} = \Omega(\boldsymbol{\theta}, \gamma) \begin{bmatrix} \mathbf{z}(t) \\ \sigma(\boldsymbol{v}(t)) \end{bmatrix}.
$$

We make the following key alterations to Eq. 24 to derive Eq. 3 for use in the policy structure:

- Remove the output equation, $y(t)$. This involves setting $D_{21}$ and $D_{22}$ to zero while replacing the linear transformation $C_2$ with our contractivity-preserving output transformation in Eq. 5.

- Eliminate the bias terms, $b_x, b_y, b_v$, as they are not essential for learning the autonomous policies we target.

- Abstract the implicit relationship between $v$ and $w$, eliminating its explicit dependency on the latent state.

### D.2  UNIVERSAL CONTRACTION

The first theorem in the paper guarantees contraction at a specific rate, contingent upon the existence of two matrix structures, as formally stated below.

**Theorem D.1.** *(Martinelli et al. (2023): Theorem 1) The REN architecture in Eq. 3 is contracting with the rate $\gamma$, if there exists a matrix $P \succ 0$ and a diagonal matrix $\Lambda \succ 0$ such that:*

$$
\begin{bmatrix} -A^\top P - PA & -C_1^\top \Lambda - PB_1 \\ * & W \end{bmatrix} \succ 0,
$$

*where,*

$$
W = 2\Lambda - \Lambda D_{11} - D_{11}^\top \Lambda.
$$

While some notations in this theorem are abstracted for simplicity in our paper, the key takeaway is that the existence of a positive-definite matrix $P$, along with a diagonal and positive-definite matrix $\Lambda$, is sufficient to ensure the contractive nature of the architecture described in Eq. 3. A detailed proof can be found in the corresponding paper, and extensive experimental results further validate this theory in practice.

## D.3 FREE PARAMETERIZATION AND CONSTRUCTABILITY

Following Theorem D.1, it is necessary to prove that there exists a way to construct the model parameter matrices, including $\Lambda$ and $P$, making the entire architecture contractive by design and with the specified contraction rate. The following theorem essentially serves that purpose. It outlines a procedure to construct all the necessary matrices to build a *contractive-by-design* REN.

**Theorem D.2.** *(Martinelli et al. (2023): Theorem 3) For any $\boldsymbol{\theta} \in \mathbb{R}^{N_\theta}$, and for any $\epsilon, \epsilon_P > 0$, there exist matrices $\{\tilde{A}, \tilde{b}\} \in \boldsymbol{\theta}$ and the matrices $(Y, W, Z, P) \in \boldsymbol{\theta}$, and a way to construct these matrices for any choice of parameters, such that:*

$$
\begin{pmatrix} -Y^\top - Y & -U - Z \\ * & W \end{pmatrix} = X^\top X + \epsilon I, \quad P = X_P^\top X_P + \epsilon_P I.
$$

*Note that $U = 0$ for autonomous models. Then, the latent state, $\mathbf{z}$, in the continuous REN architecture defined by Eq. 3:*

$$
\begin{bmatrix} \dot{\mathbf{z}}(t) \\ \boldsymbol{v}(t) \end{bmatrix} = \Omega(\boldsymbol{\theta}, \gamma) \begin{bmatrix} \mathbf{z}(t) \\ \sigma\big(\boldsymbol{v}(t)\big) \end{bmatrix},
$$

*is contractive with the rate $\gamma$ for any choice of $\boldsymbol{\theta}$. Further, matrices $(Y, W, Z, P)$ and $\{\tilde{A}, \tilde{b}\}$ can be computed unambiguously as described in the proof.*

The full proof for this theorem may be found in Martinelli et al. (2023), p. 8. All the matrices required in the theorem can be efficiently designed and computed using standard automatic differentiation tools, such as PyTorch, so that the resulting model is trainable, just like a normal recurrent neural network.

# E  ADDITIONAL EXPERIMENTS

We present a set of additional experiments that highlight various features of SCDS, alongside ablation studies on its key components: contractive RENs and the output map.

## E.1  FURTHER EXPERIMENTS ON THE LASA DATASET

We illustrated in Sec. 5 how the state trajectories contract towards each other. As explained in Subsec. 2.3, the latent state trajectories, $\mathbf{z}$, also exhibit contraction. To illustrate this, we trained a policy with different contraction rates for a sample motion from the LASA dataset and plotted the first element of the latent state when starting from various initial conditions in Fig. 8. In this experiment, the latent state has a dimension of 10. As expected, the latent states contract toward each other, with the contraction speed increasing for larger values of the contraction rate $\gamma$.

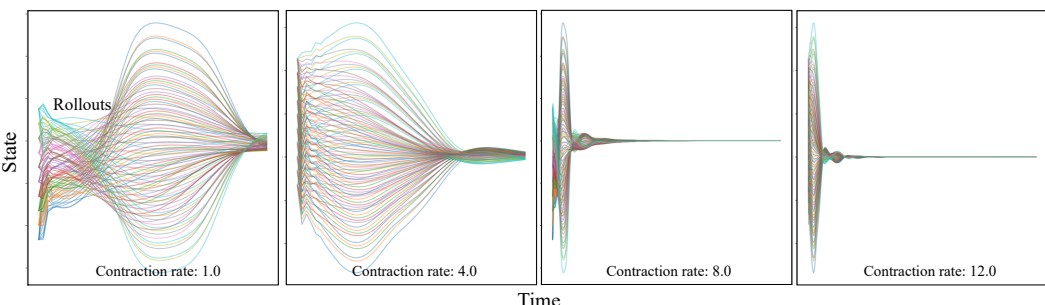

Figure 8: Contractive behavior of the latent state. The figure illustrates the rollouts over time for varying dimensions of the latent state, under different contraction rates. Higher contraction rates accelerate convergence in both the latent state space and the robot's state space. The dimension of the latent state here is set to 10.

In Fig. 9, we complement our experiments in Sec. 5 on the LASA dataset by illustrating more expert motions. For all motions, SCDS produces trajectories that contract to an average of expert demonstrations, resulting in a high imitation accuracy.

Next, we analyze the effect of the distance of an OOS initial state from those in $\mathcal{D}$ in Fig. 10 and Tab. 3. As can be seen in the figure, SCDS demonstrates effective OOS recovery, even when the initial states are very far from the in-sample ones.

Table 3: Evaluating out-of-sample rollouts error on the LASA dataset. Compared to Table 1, the OOS initial states are sampled from a wider distribution with a radius of $0.25\|\mathbf{y}_0^m\|_2$ for the LASA dataset. The lowest (best) values for each metric and dataset are highlighted. These results confirm that the recovery behavior observed in Fig. 10 persists across all the other tasks in the LASA dataset.

| Expert | LASA-2D | | LASA-4D | |
|---|---|---|---|---|
| Metric | MSE | soft-DTW | MSE | soft-DTW |
| SNDS | $4.64 \pm 2.50$ | $11.06 \pm 2.35$ | $6.57 \pm 3.56$ | $14.78 \pm 1.08$ |
| BC | $13.81 \pm 6.08$ | $29.25 \pm 8.45$ | $32.73 \pm 11.45$ | $41.22 \pm 11.51$ |
| SDS-EF | $2.67 \pm 0.55$ | $12.12 \pm 2.29$ | $4.19 \pm 0.78$ | $16.34 \pm 3.28$ |
| SCDS | $0.57 \pm 0.23$ | $2.12 \pm 0.70$ | $1.44 \pm 0.35$ | $3.40 \pm 0.56$ |

## E.2  LEARNING THE CONTRACTION RATE

As discussed in App. D and Eq. 3, we can either set the contraction rate $\gamma$ to a desired value (see Tab. 7) or learn it as a model parameter. For learning $\gamma$, the Lagrange multipliers method can be used which replaces the loss $\hat{\mathcal{L}}$ with:

$$\hat{\mathcal{L}}_{\mathrm{aug}}\left(\left\{\mathbf{y}_0^m\right\}_{m=1}^M; \boldsymbol{\theta}, \gamma\right) \triangleq \hat{\mathcal{L}}\left(\left\{\mathbf{y}_0^m\right\}_{m=1}^M; \boldsymbol{\theta}\right) - \mu\left(h(\gamma) - c\right), \tag{26}$$

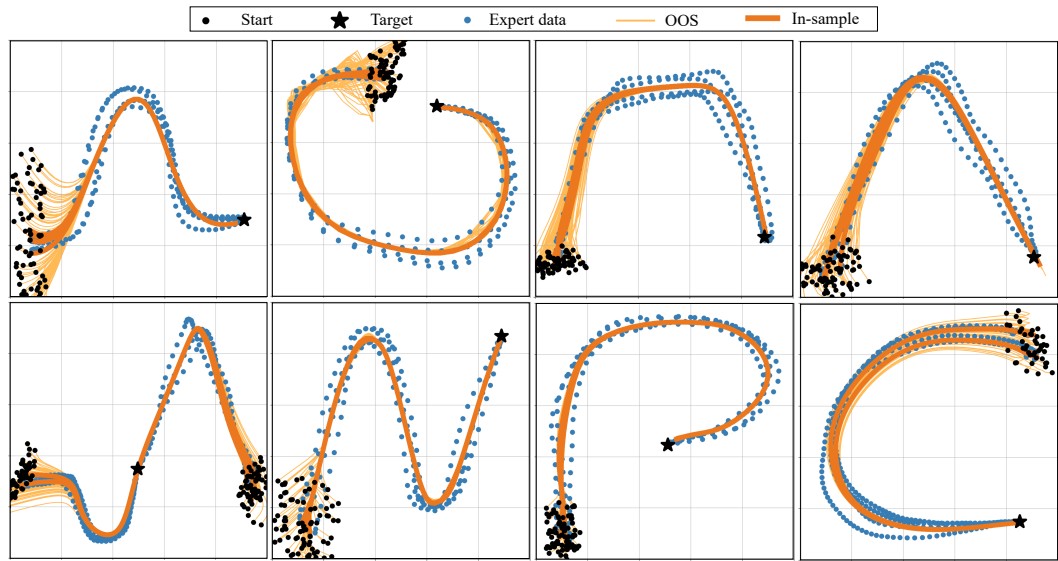

Figure 9: SCDS is tested over additional motions in the LASA dataset. In all cases, the induced trajectories accurately imitate the expert, even for out-of-sample initial states.

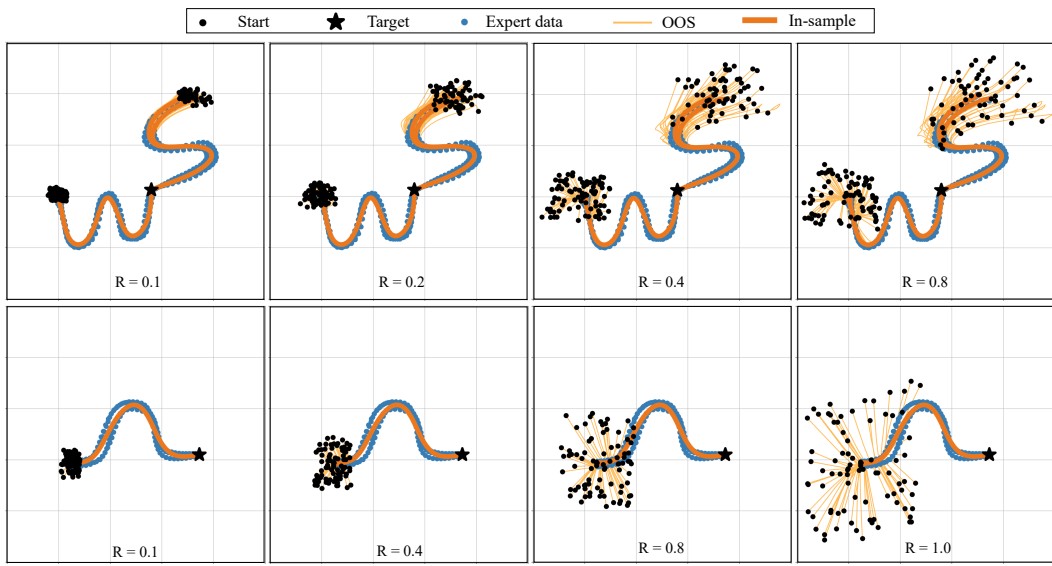

Figure 10: Out-of-sample recovery for initial states sampled from regions with different volumes according to the definition of $R$ in Assumption 4.1. Notably, SCDS recovers even from more distant initial states.

where $h(.)$ can be defined to encourage larger $\gamma$, and the constants $\mu$ and $c$ are regularization terms. A natural choice for $h$ can be $h(\gamma) = \frac{1}{(\gamma - \gamma_0)^2}$, with $\gamma_0$ being a baseline value.

Fig. 11 illustrates the difference in out-of-sample recovery between two cases: setting a low contraction rate (top row) versus promoting higher contraction rates through Eq. 26 (bottom row). As shown, learning $\gamma$ leads to a higher contraction rate, which results in trajectories that contract toward each other more quickly.

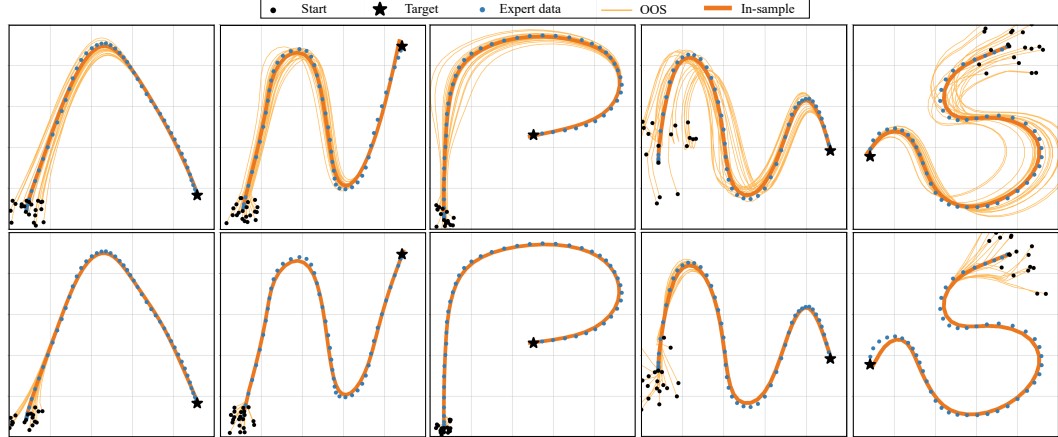

Figure 11: Policies trained to have low (first row) and high (second row) contraction rates. SCDS can promote higher contraction rates during training to achieve more robustness to out-of-sample initial states and perturbations.

### E.3 LATENT STATE DIMENSION

One of the key advantages of SCDS is its ability to learn in a latent space. As discussed in Sec. 1, increasing the latent space dimension enhances the expressive power of the policy. We explore this aspect in Fig. 12 for a 2D motion in the LASA dataset. As shown, a low-dimensional latent space restricts representation capacity, leading to poor imitation. However, as the latent dimension increases, imitation accuracy improves.

It is important to highlight that the policy remains contractive by design, regardless of the latent state dimension. This ensures that even the poorly trained policy in the left-most plot of Fig. 12 remains globally stable and contracting.

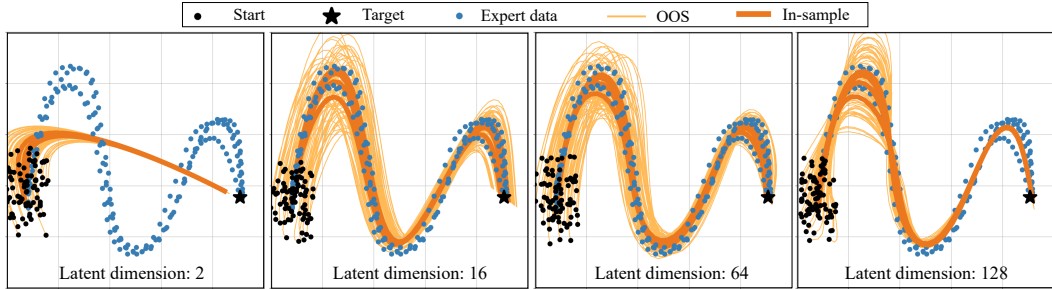

Figure 12: Training results for a sample motion in the LASA dataset are plotted for various latent space dimensions. Training in higher dimensional latent space is more effective, especially when facing complex expert behavior that involves multiple demonstrations. Note that the time horizon is fixed here for all the rollouts.

### E.4 ABLATION STUDY: BIJECTIVE LAYERS

As discussed in Subsec. 2.3, bijective layers efficiently enhance the expressive power of the policy. To empirically validate this, we compare the in-sample loss, which is an indicator of the policy's expressiveness, in two cases: first, training a policy with bijection layers (referred to as SCDS) and second, training a policy without bijection layers (referred to as SCDS-NB). The REN structure remains identical across both SCDS and SCDS-NB. The results for the LASA and Robomimic

datasets are presented in the first two rows of Tab. 4. As expected, SCDS consistently and significantly outperforms SCDS-NB.

However, directly comparing SCDS and SCDS-NB is not entirely fair, as SCDS includes more parameters and, therefore, is expected to have greater flexibility. To address this, we introduce a third policy, SCDS-NB*, which lacks bijection layers but compensates by having a more complex REN structure, achieved by increasing the size of the implicit layer, $N_v$ (see Subsec. 2.2). We show in Tab. 4 that SCDS and SCDS-NB* achieve comparable performance. However, it is important to note that training SCDS is significantly faster than training SCDS-NB*. This finding demonstrates that using bijection layers is a more efficient alternative to complicating the REN structure when seeking to increase flexibility.

Table 4: Ablation study on invertible layers to improve the policy's expressive power. SCDS is the standard policy formulation presented in Eq. 7. SCDS-NB uses the same REN structure but does not have bijection layers. SCDS-NB* makes SCDS-NB more complex by increasing the complexity of the implicit layer in the REN architecture. Still SCDS-NB* uses no invertible layers. The best results are displayed for 50 trials on in-sample initial states.

| Expert | LASA-2D | | LASA-4D | | Robomimic-6D | | Robomimic-14D | |
|---|---|---|---|---|---|---|---|---|
| Metric | MSE | soft-DTW | MSE | soft-DTW | MSE | soft-DTW | MSE | soft-DTW |
| SCDS | $0.02 \pm 0.01$ | $0.65 \pm 0.05$ | $0.03 \pm 0.01$ | $0.72 \pm 0.12$ | $0.56 \pm 0.22$ | $1.05 \pm 0.37$ | $1.68 \pm 0.45$ | $4.10 \pm 0.40$ |
| SCDS-NB | $0.43 \pm 0.18$ | $2.41 \pm 0.75$ | $1.45 \pm 0.30$ | $3.38 \pm 0.42$ | $1.24 \pm 0.35$ | $2.23 \pm 0.74$ | $3.49 \pm 0.85$ | $8.24 \pm 0.64$ |
| SCDS-NB* | $0.01 \pm 0.00$ | $0.50 \pm 0.06$ | $0.04 \pm 0.02$ | $0.79 \pm 0.13$ | $0.45 \pm 0.15$ | $0.88 \pm 0.26$ | $1.81 \pm 0.24$ | $4.87 \pm 0.32$ |

### E.5 ABLATION STUDY: REPLACING REN WITH A NON-CONTRACTIVE MODEL

In this section, we analyze the impact of the contractivity of REN on the resulting policy. To this end, we replace the REN in Eq. 7 with a recurrent neural network (RNN), a subclass of REN that is not contractive. Although the RNN architecture captures temporal dependencies, OOS trajectories are not guaranteed to converge toward expert data, as the resulting policy is no longer contractive.

We illustrate this behavior in Fig. 13 for both the LASA and the Robomimic datasets. Furthermore, Fig. 14 plots the average distance between trajectories for SCDS and the non-contractive baseline over time. Both figures highlight that some OOS trajectories diverge when using the RNN, while all trajectories consistently converge to the target when using the contractive REN. This analysis reinforces the intuition behind employing contractive RENs as the foundation of the policy.

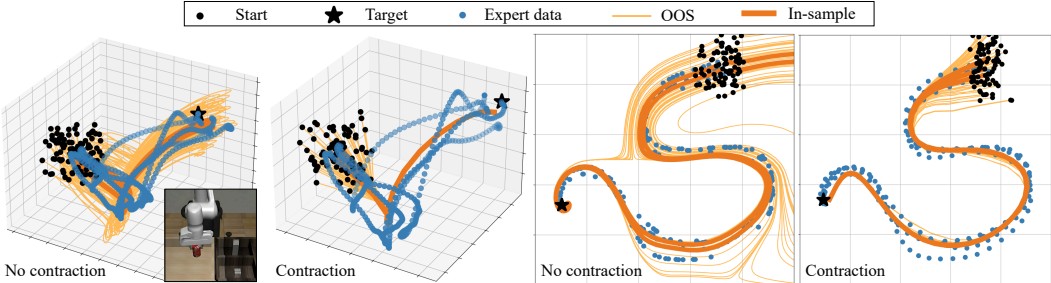

Figure 13: The effect of replacing the contractive REN with a non-contractive model, specifically a recurrent neural network, is investigated across two candidate tasks. Some of the generated trajectories diverge in the *no contraction* case, which lacks guarantees on out-of-sample recovery.

### E.6 COMPUTATION TIME

We compare the training time of SCDS with the employed baselines in Tab. 5, using the same number of epochs and data points. The hyperparameters employed in this comparison are optimized for the best accuracy for each baseline, as detailed in Tab. 6 and Tab. 7. Despite the longer training

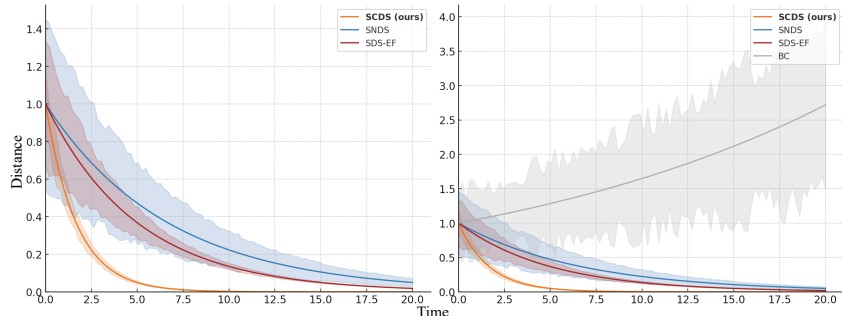

Figure 14: Distance between induced trajectories for the policies using a contractive REN (left) and a non-contractive RNN (right), with a uniformly distributed set of initial states around the true initial conditions from the LASA dataset. Since the policy in the right figure is not contractive or stable, the trajectories begin to diverge, causing the distance between neighboring trajectories to increase.

times, our approach ensures contraction, offering a stronger stability guarantee compared to the weaker stability guarantees provided by the stable baseline methods, or no guarantees at all in the case of BC.

It is important to note that, since our method relies solely on state data rather than state time derivatives, the computational cost can be significantly higher in lower-dimensional settings. However, this disparity decreases in higher dimensions, particularly when utilizing parallel computation for the generated trajectories. Additionally, switching to a discrete REN architecture can notably reduce computation time, though at the expense of increased memory consumption.

Table 5: Evaluating the training time on the LASA and the Robomimic datasets in seconds. SCDS using either discrete or continuous REN architecture appears to be less computationally efficient in lower dimensions, but will shorten the gap in efficiency in larger datasets. Note that even though the same number of epochs are used here, the continuous SCDS method typically requires less than half of the number of epochs to achieve the same performance. Although SCDS has longer training times, it offers the stronger contractivity guarantee compared to the stability guarantees of the stable baselines or the lack of any guarantees in the case of BC.

| Expert | LASA-2D | LASA-4D | Robomimic-6D | Robomimic-14D |
|---|---|---|---|---|
| SNDS | $156.8 \pm 14.1$ | $372.5 \pm 22.8$ | $435.2 \pm 17.9$ | $882.2 \pm 45.7$ |
| BC | $57.2 \pm 10.1$ | $62.9 \pm 12.5$ | $91.5 \pm 17.4$ | $103.0 \pm 36.5$ |
| SDS-EF | $417.1 \pm 18.7$ | $658.8 \pm 27.2$ | $825.7 \pm 59.9$ | $1150.2 \pm 94.5$ |
| **SCDS-Continuous** | $1794.7 \pm 92.8$ | $1767.2 \pm 94.6$ | $2190.2 \pm 43.2$ | $2711.1 \pm 173.5$ |
| **SCDS-Discrete** | $381.9 \pm 32.6$ | $509.0 \pm 18.2$ | $610.5 \pm 36.3$ | $857.4 \pm 64.6$ |

### E.7 LONGER HORIZON MOTIONS

In this section, we use the Snake expert demonstrations employed for experiments in LPV-DS (Figueroa & Billard, 2018) and SNDS (Abyaneh et al., 2024). The Snake motion provides a stronger challenge than the typical tasks in the LASA dataset. Fig. 15 compares SCDS results against the best result mentioned in SNDS. Numerically, the average DTW for SNDS is reported as roughly $0.158$. However, SCDS results in a better imitation accuracy, with an average DTW error of $0.074$ with a low contraction rate, and $0.097$ with high contraction rates as depicted in Fig. 15.

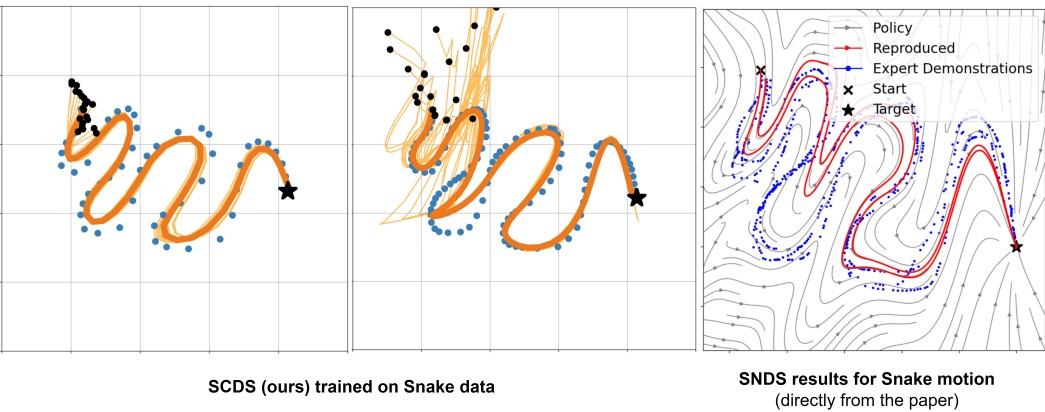

**SCDS (ours) trained on Snake data**     **SNDS results for Snake motion**
(directly from the paper)

Figure 15: SCDS results for the Snake dataset demonstrate its ability to handle a more complex motion compared to the tasks in the LASA dataset. SCDS is applied starting from closer (left plot) and further (middle plot) initial conditions and is compared against SNDS (right plot). The complexity of the Snake task arises from the Snake dataset's longer time horizon and the need for rapid adaptation in DS learning.

## F   Implementation details

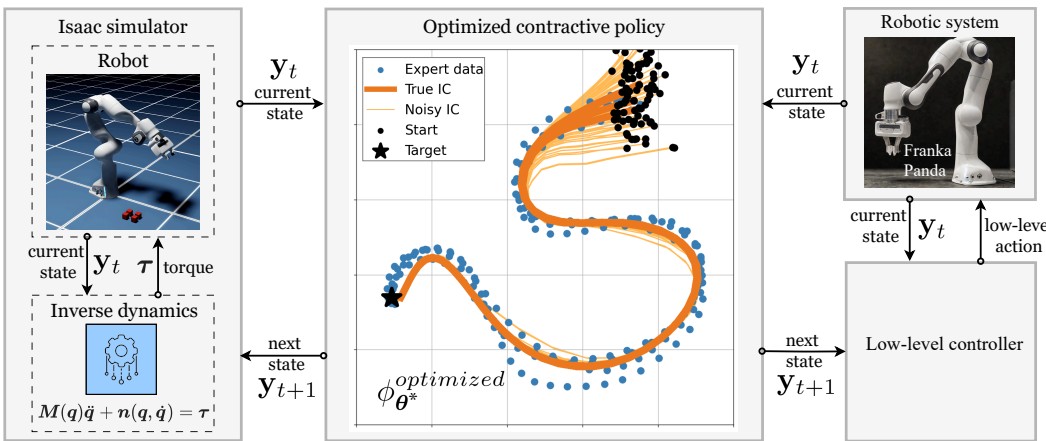

Figure 16: Simulation and real-world deployment pipelines. The trained policy can be deployed to both real and simulated robots with minimal effort. Global stability and highly contractive behavior of induced trajectories ensure safety and precision during these zero-shot transfers.

This section is focused on key components of SCDS's efficient implementation, hyperparameters, and further details about our selected baselines and their architecture. Note that the entirety of SCDS's codebase, including the following modules, are efficiently implemented in PyTorch (Paszke et al., 2019).

### F.1   Key components of SCDS

As discussed in Sec. 2, our approach benefits from three major design choices to enable (1) contractivity by design and with an adjustable contraction rate, (2) additional expressivity, and (3) efficiently imitating the expert's behavior.

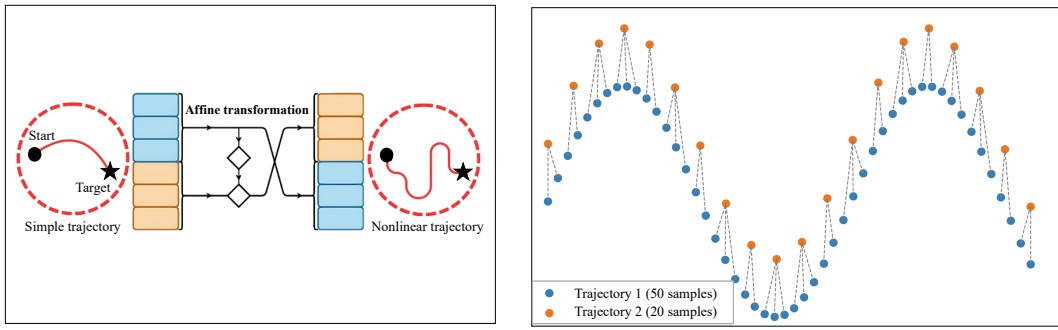

Figure 17: Left: Invertible coupling layer structure implemented as a part of RealNVP. The affine transformation provides increased nonlinearity while maintaining bijection, and as a result, contraction properties. Right: DTW still effectively compares trajectories with different lengths. This is particularly useful when learning long-horizon trajectories, or dealing with higher sampling rates.

**(1) Continuous and discrete RENs**   There are two implementations of the recurrent equilibrium network architecture: the Martinelli et al. (2023)'s continuous time [5] approach, and the original

---

[5] github.com/DecodEPFL/NodeREN

Julia implementation of Revay et al. (2023)'s discrete-time method [6] formulation. Our repository supports both discrete and continuous REN architectures, though most of our experiments focus on continuous RENs. The discrete-time approach can be computationally more efficient, particularly over long horizons. However, it is less memory efficient, as the gradient tree expands with the horizon length.

In contrast, continuous RENs are compatible with Neural ODEs and still induce well-posed and contractive dynamical systems. Additionally, their compatibility with various integration schemes provides a degree of freedom in balancing accuracy and computational cost, as well as the ability to evaluate at arbitrary time points without the need for uniform time sampling (Martinelli et al., 2023). Hence, the policy learned by using continuous RENs outputs smoother generated trajectories with adjustable granularity.

**(2) Invertible coupling layers**  The RealNVP implementation of coupling layers (Dinh et al., 2017) is a key mechanism for constructing invertible transformations while maintaining computational efficiency. These layers split the input data into two parts, where one part remains unchanged,

and the other is transformed using a function parameterized by the unchanged portion. Specifically, the transformed part is updated via an affine transformation, where the *scale* and *shift* parameters depend on the static part. This ensures that the transformation remains invertible, as the unchanged part and the inverse of the affine transformation can be easily recovered. The structure of coupling layers allows for efficient computation of both the forward pass and the log-determinant of the Jacobian, which is crucial for likelihood estimation in normalizing flows.

**(3) Soft and differentiable dynamic time warping**  Dynamic Time Warping (DTW) (Keogh & Ratanamahatana, 2005) and its faster variant, soft-DTW (Cuturi & Blondel, 2017), are often used to measure similarity between curves by aligning sequences that may vary in speed, timing, or sampling rate. However, DTW can be computationally expensive, which is where soft-DTW becomes useful. Soft-DTW offers a more efficient approximation with adjustable reduced complexity while maintaining relatively high accuracy.

DTW methods outperform MSE in scenarios where temporal variations or differences in sampling rate are key to understanding the similarity between expert behavior and generated rollouts.

## F.2 BASELINES

This section contains information about the baselines, especially their codebase and parameters, summarized in Tab. 6. Note that we were unable to compare our method against more recent contractive methods, especially Mohammadi et al. (2024), as there is no official implementation released with the paper.

**SDS-EF**  encodes expert demonstrations as dynamical system trajectories on a Riemannian manifold, then transforms them into simpler straight-line paths in a latent Euclidean space using diffeomorphisms (Rana et al., 2020). The *gradient flow* dynamical system is designed so that trajectories in the latent space are linear and converge towards a stable equilibrium. By ensuring that the diffeomorphic transformation used to map the original Riemannian manifold to this Euclidean space is smooth and invertible, the stability of the trajectories is maintained when mapping back to the original space with theoretical guarantees. Hence, SDS-EF can learn and reproduce complex expert data while providing asymptotic stability.

**SNDS**  guarantees global asymptotic stability by jointly training neural networks for both the policy and a trainable Lyapunov candidate (Abyaneh et al., 2024). Therefore, it ensures trajectories converge predictably even under perturbations or out-of-sample initial conditions. SNDS is tested on high-dimensional robotic tasks, and its differentiable loss aligns policy rollouts with expert demonstrations. The method shows strong performance in accuracy and stability across complex environments, such as robotic arm control tasks, but suffers from non-smoothness with rapidly changing expert behavior.

---

[6] github.com/acfr/RobustNeuralNetworks.jl

**BC** is the standard approach to learning from expert demonstrations without additional safety or stability guarantees. BC-RNN (Mandlekar et al., 2020) is a variation of the original behavioral cloning, which uses recurrent structures to capture emerging patterns over time. Nonetheless, we still employ a vanilla BC implementation, as the experiments are not focused on long-horizon tasks with substantial recurring patterns.

Table 6: Baseline implementation, architecture, and hyperparameters

| Baseline implementations | | |
|---|---|---|
| **Method** | **Codebase** | **Stability certificate** |
| SNDS (Abyaneh et al., 2024) | github.com/aminabyaneh/stable-imitation-policy | Lyapunov, *exponential, asymptotic* |
| SDS-EF (Rana et al., 2020) | github.com/mrana6/euclideanizing_flows | Lyapunov, *asymptotic* |
| BC (Pomerleau, 1988; Mandlekar et al., 2020) | github.com/montaserFath/BCO | None |
| **Baseline parameters** | | |
| SNDS | SDS-EF | BC |
| *policy* network: [2, 256, 256, 256, 2] | bijection blocks: 10 | network: [2, 256, 256, 256, 2] |
| *ICNN* network: [2, 64, 64, 1] | hidden layers: 100 | type: *ResNet* or *FeedForward* |
| activation: *nn.SoftPlus* | activation: nn.ELU | activation: *nn.ReLU* |
| $\alpha = 0.01$ | coupling layer: 'rfnn' | regularization: $1e-5$ |
| $\epsilon = 0.01$ | $\epsilon = 1e-5$ | |

Table 7: Hyperparameters of SCDS and their optimal values or ranges are shown for every key model parameter.

| Param | Description | Optimal value(s) |
|---|---|---|
| $\gamma$ | Lower bound on the contraction rate of the policy | $[1.0, 18.6]$, *learnable* |
| $K$ | Number of coupling layer to increase the output nonlinearity | 4 to 10 |
| $N_{\mathbf{z}}$ | Dimension of the latent state space | 32 to 64 |
| $H$ | Forward simulation horizon for the generated trajectory | 20 to 50 |
| $K$ | Number of invertible layers in the output map (also used for hidden blocks) | 4, 8 |
| $\beta$ | Complexity-accuracy trade-off parameter for soft-DTW ($\gamma$ in some references) | 0.1 |

### F.3 COMPUTATIONAL RESOURCES

For our experiments, we relied on a computational server running on the Linux operating system, specifically Ubuntu 24.04.1 LTS. The server was equipped with a high-performance NVIDIA RTX 4090 GPU with CUDA drivers version 12.6, which are more efficient when it comes to optimizing a loss on several rollouts generated in parallel. Moreover, an array of CPUs, Intel Core i9-9900K, featuring 8 cores and 16 threads each, and 64 GB of DDR4 RAM (2x32 GB), were used in conjunction with the single GPU pipeline.

To conduct simulations with Isaac Lab for robotics experiments, we required 4-6 GB of VRAM and 4-8 GB of system RAM, depending on the simulation task. We ran the simulations remotely on the server cluster. To interact with these simulations, we used the Omniverse Streaming Client, allowing us to connect to the server and receive real-time broadcasts of the experimental scenes. This setup ensured that we could handle the demanding requirements of our simulations while maintaining low latency and high-quality visualization during the remote experiments. Note that Isaac Lab enables parallel training and testing across randomized environments, as depicted in Fig. 18.

### F.4 SIMULATION IN ISAAC LAB

Isaac Lab [7] is a cutting-edge framework built on top of NVIDIA's Isaac Sim [8], designed for creating and simulating complex robotics systems. It leverages the high-performance computational power of NVIDIA GPUs to efficiently simulate and train robots in a realistic virtual environment. By

---

[7] https://isaac-sim.github.io/IsaacLab/index.html
[8] https://developer.nvidia.com/isaac/sim

integrating Isaac Sim's physics-based simulation with Isaac Lab's scalable infrastructure, we can run multiple parallel environments.

As shown in Fig. 18, this level of parallelism greatly reduces both training and testing time, the latter being particularly critical for our application. SCDS is integrated into robotic systems using the workflow described in Fig. 16. The low-level controller translates contractive policies in the joint and task space of the robot into low-level joint torque or velocity commands, depending on the specific application. It is important to note that no low-level controller is necessary for joint position control, as the policy directly produces the next state. For task space position and orientation, only a differential kinematics approach is needed to convert the end-effector pose into joint space.

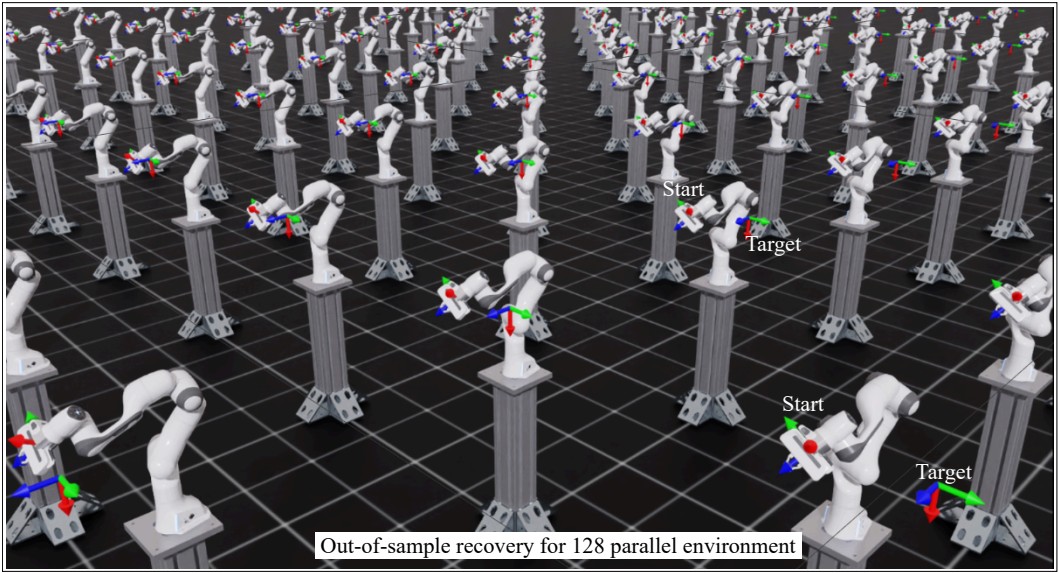

Figure 18: Efficient parallel testing of out-of-sample behavior in the Isaac Lab robotics simulator. Each robot is given a random initial condition drawn from a uniform distribution around true initial states. The parallel testing of 128 sampled initial conditions yields the results in Tab. 1.

### F.5 LOW-LEVEL CONTROLLER

Throughout this paper, we have assumed that there exists a low-level controller that converts the state planned by the DS $\phi_\theta$ into motor commands (e.g., force or torque). This is a standard assumption in learning high-level actions in imitation learning (Khansari-Zadeh & Billard, 2011; Torabi et al., 2018; Chi et al., 2023), reinforcement learning (Bahl et al., 2020), and optimal control (Boroujeni et al., 2021). In this section, we discuss designing this low-level controller.

In our policy formulation, the high-level part of the policy takes the current state of the robot to the next desired state. This state representation can be defined either in the joint space as in our experiments with the Franka Panda arm in Fig. 7, or in the robot's task space as in the experiments with Clearpath Jackal in Fig. 7.

Once the high-level policy determines the next desired state, this information is relayed to a low-level controller responsible for generating low-level torque or velocity commands. Depending on the specific requirements of the task and the robot's configuration, we utilize one of three types of controllers: a joint position controller, an inverse dynamics controller, or an inverse kinematics controller. The joint position controller directly commands the robot's joints to move to the specified positions. The inverse dynamics controller calculates the necessary torques or forces at each joint to achieve the desired motion. The inverse kinematics controller computes the joint configurations needed to attain a specific end-effector position and orientation. This hierarchical control strategy enables precise and efficient motion control by leveraging the strengths of both high-level planning and low-level execution mechanisms.

# G    DATASETS

We employ two well-known datasets to evaluate our method on real-world expert trajectories. The LASA dataset Khansari-Zadeh & Billard (2011) has long been the primary benchmark for learning DS-based policies, so we begin our experiments with this standard dataset. However, we do not stop after obtaining promising results, as SCDS demonstrates capabilities beyond the 2-dimensional and 4-dimensional expert demonstrations in the LASA dataset. To further challenge our method, we utilize the Robomimic dataset Mandlekar et al. (2021) for the first time in the literature on learning policies modeled by DSs, allowing us to evaluate SCDS in more complex 6-dimensional and 14-dimensional spaces.

## G.1    LASA DATASET

The LASA Handwriting Dataset is a comprehensive collection of 2D handwriting motions recorded from a human expert drawing on a tablet. It is particularly useful for visualization, and learning semi-complex dynamics. The dataset contains 30 distinct handwriting motions. Each motion is represented by 7 demonstrations, with the user drawing the desired pattern starting from different initial positions and ending at the same final point, typically with some intersection between the demonstrations. The handwriting motions are defined in a 2D Cartesian space, with the final target set at coordinates (0, 0) for all patterns, without loss of generality. The dataset provides consistent 1000 data points per demonstration, interpolated to handle variations in timing between different demonstrations. The data structure provides insights into both the spatial and temporal dynamics of handwriting patterns, including the position, velocity, and acceleration of each motion.

Despite having 1000 samples per trajectory, the expert behavior may be replicated with very high accuracy using a limited horizon $H$ of 50 to 100. Since SCDS employs the soft-DTW loss, which does not require the trajectories to have the same length, we can efficiently learn every motion in the LASA dataset while keeping the computational complexity to a minimum.

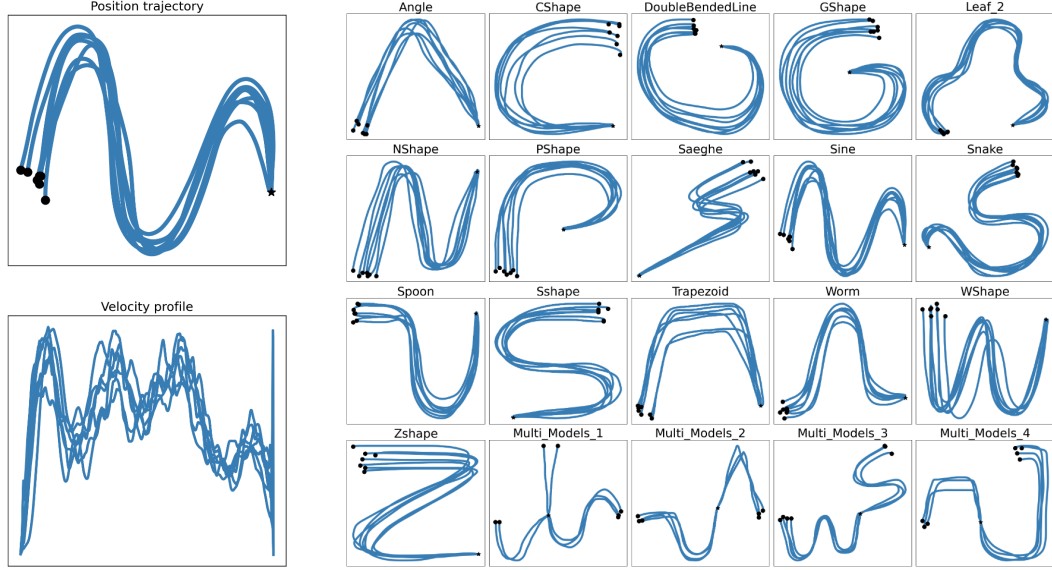

Figure 19: Expert demonstrations in the LASA dataset.

## G.2    ROBOMIMIC DATASET

The Robomimic dataset is a complete benchmark for robot learning from offline human demonstrations, designed to cover a wide range of robot manipulation tasks. The dataset includes data gathered from experts performing a variety of tasks through teleoperation, and also artificial data generated

by policies which are optimized using well-established reinforcement learning techniques. We focus on the data generated by human experts for a set of designated tasks, namely, "Lift", "Can", "Square", and "Transport". The Robomimic dataset provides detailed measurements of the observation space. Most notably, the dataset features are the robot's joint position (7D), joint velocity (7D), end-effector position (3D), end-effector orientation (3D), etc. We combine these features to get higher dimensional augmented state spaces (14D in the joint space, 6D in task space) to better evaluate SCDS and the selected baselines described in App. F.2.

The expert demonstrations for these tasks are depicted in Fig. 20. The Lift and Can tasks are relatively easier to perform than the more complicated Square and Transport tasks. Notably, the average number of samples per task is 48 ± 6 (Lift), 116 ± 14 (Can), 151 ± 20 (Square), and 469 ± 54 (Transport). Generally, we observe that the more complicated a task in this dataset, the more samples are required to describe it. There are between 200 and 300 demonstrations per task, but we typically restrict the learning to a dominant subset (between 80-90 percent) of these demonstrations to train SCDS and leave the rest to the evaluation stage.

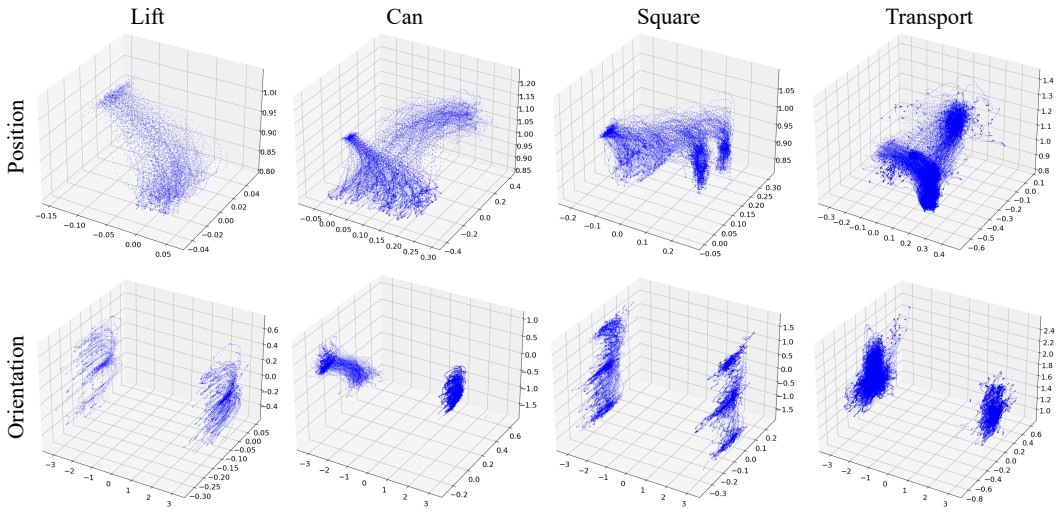

Figure 20: Position and orientation data for different tasks in the Robomimic dataset.

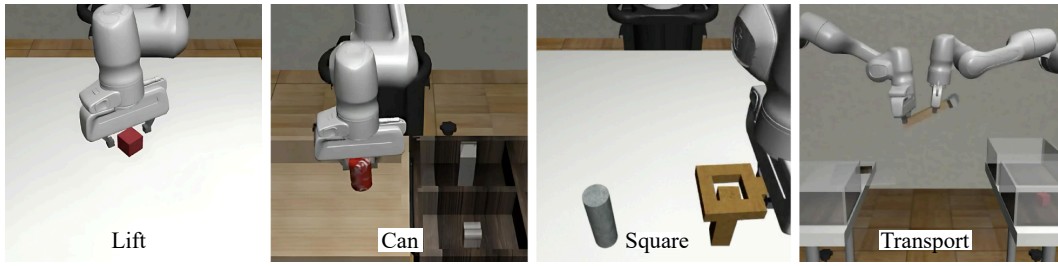

Figure 21: Data collection setup for Lift, Can, Square, and transport tasks (Mandlekar et al., 2021). We employ this collected data, which is available in the Robomimic repository.

