# OpenReview forum: "Contractive Dynamical Imitation Policies for Efficient Out-of-Sample Recovery"
_ICLR.cc/2025/Conference — ICLR 2025 Poster_

### Official Review · Reviewer_hPP6 · 2024-10-29

**Soundness:** 2
**Presentation:** 3
**Contribution:** 2
**Rating:** 6
**Confidence:** 2

**Summary:**

This paper studies how to learn a policy with contractive guarantee for out-of-sample recovery on a dynamical system. Remarkably, The paper 1) instantiates their DS policy through contractive REN, bijective transformation and neural ODE solver, with careful considerations of their each feature and applicability; and 2) provides thorough mathematical analysis on contractivity of the  proposed methods and an upper-bound of the loss when initial states locate in a multi-focal ellipse region. Finally, the authors provide empirical results on 2 datasets and a simulated environment, demonstrating the favorable performance of their methods.

**Strengths:**

## originality:
The paper creatively integrates several interesting techniques, i.e. contractive REN, neural ODE and coupling layers into their methods, resulting in a successful and analyzable performance. To my knowledge, the method is both novel and effective, with a clean and elegant design.

## quality:
Some mathematical analysis are performed on the contractivity of the proposed method, SCDS. Notably, an upper-bound of the true loss $\mathcal{L}(\rho_0; \theta)$ is derived and provides much insights.

## clarity:
The paper is well written with clean words and appealing figures. It is easy to read and comprehend.

## significance:
Under the condition considered (the initial state locates in a multi-focal ellipse of in-data samples), the method proposed offers rigorous guarantee of contractivity and favorable  performance, which is illustrated by numerical and visual results in experiment section.

**Weaknesses:**

I have 2 major concerns of this work:

1.  the lack of consideration of  control input $u$, and the system dynamics with it, $\dot{s} = f(s, u)$. Together with the ignorance of first-derivative of the system state. The author's  problem formulation reduces to how to generate state-only trajectory to meet the desired system requirement. Unfortunately, in my opinion, this approach of system modeling and problem formulation doesn't tackle the difficulty we usually meet in Robotics: 1) the real-world robotic system always has dynamics capacity and not every planned trajectory is equally executable hence the real-world impact of SCDS remains unknown; 2) Without the consideration of system dynamics in terms of control input, it's hard to assess the significance of the proposed method. Actually, under this simplified situation, the theoretical derivations and contractivity guarantees seem to be an over-kill.  One could simply generate trajectory interpolating towards the nearest in-sample point then exactly follow its trajectory.

2. the overly-simplified experiment settings. Besides the simplification in problem formulation, the experiment settings are also too narrow. There are 2 settings of the initial states in the experiments: 1) the initial states are sampled directly from the dataset; and 2) the initial states are sampled from a multi-focal ellipse of the in-dataset samples, with a radius of 0.1 of its norm. I think this experiment setting is neither challenging nor fitting for the real-world cases where OOS recovery is in great need.

**Questions:**

for concern 1,

1. could you provide papers/references to showcase how your problem formulation could lead to  methods with successful real-world deployments/applications on robotic system?
2. could you provide analysis on the realizability of the trajectory generated by your methods? i.e. does it always generate trajectory that is executable for the underlying robotic system? if not, could you provide analysis or experiments on the performance of your method under this situation?

for concern 2,

1. In your experiments, could you set the initial states to be in some regions where state visitation density is low and show its performance?
2. could you try training a diffusion model based policy (with treating the problem as future state generation conditioned on current state and train it with a data augmentation that adds the same level of ellipse noise to conditioned input state) and evaluating it in your experiments? I suspect this would already generate a decent performance on the experiments you conducted.

---

> ### Author Response · Authors · 2024-11-18
> **Response to reviewer's comments**
>
> We sincerely thank the reviewer for their valuable suggestions. Each comment has been addressed in the sequence provided, and we have incorporated additional improvements throughout the manuscript. An updated version of the manuscript is released.
>
> **1. Papers/references showcasing successful real-world deployments/applications on robotic systems**
>
> Learning dynamical systems has long-established literature, and many works have demonstrated real-world applications. Below are a few papers demonstrating successful deployment for real-world sensorimotor learning:
>
> (1) Rana, Muhammad Asif, et al. "Euclideanizing flows: Diffeomorphic reduction for learning stable dynamical systems." Learning for Dynamics and Control. PMLR, 2020.
>
> (2) Mohammadi, Hadi Beik, et al. "Neural contractive dynamical systems." The Twelfth International Conference on Learning Representations, 2024.
>
> (3) Abyaneh, Amin, Mariana Sosa Guzmán, and Hsiu-Chin Lin. "Globally Stable Neural Imitation Policies." In IEEE International Conference on Robotics and Automation (ICRA), 2024.
>
> **2. Realizability of the trajectory generated by the proposed methods**
>
> Assessing the feasibility of the induced trajectories is a major challenge in model-free imitation learning. In general, imitation learning methods cannot fully address trajectory feasibility for out-of-distribution regions of the state space without additional mechanisms. Best mitigations typically involve data augmentation or expert-in-the-loop setups, as discussed in [1, 2].
> In our case, while our method guarantees contractive recovery from out-of-distribution states, the feasibility of the generated trajectories cannot be ensured without access to additional data or a dynamics model of the robot. However, both these requirements conflict with the assumptions of our model-free and offline approach. In many practical tasks, trajectory feasibility is mitigated by passing the next state to a low-level controller, such as a model predictive controller, which generates collision-free torque or velocity commands for the robot. While this addresses the feasibility problem at the lower level, the limitation could persist at the planning level.
>
>
> **3. Setting the initial states in regions with a low state visitation density**
>
> We would like to highlight that Figure 10 in the original submission analyzes the impact of the spread of initial conditions using a task from the LASA dataset. For further details, please refer to our answer 9 to reviewer AXT4.
>
> **4. Could you try training a diffusion model based policy (with treating the problem as future state generation conditioned on current state and train it with a data augmentation that adds the same level of ellipse noise to conditioned input state) and evaluating it in your experiments? I suspect this would already generate a decent performance on the experiments you conducted.**
>
> While the representational power of diffusion-based policies has been demonstrated in recent work [3], these methods typically treat the diffusion process as a black box to model action distributions. As the reviewer points out, training on augmented data might yield desirable outcomes, but there are no theoretical guarantees for samples outside the augmented dataset. This highlights a key difference between black-box learning approaches, like diffusion policies, and our method, which is based on dynamical systems [4] with theoretical stability guarantees. In our approach, contractivity is ensured without requiring augmented data, even for out-of-sample initial states far from the training ones.
> Additionally, we note that diffusion-based policies generally rely on significantly more parameters (approximately 6.5e+07 [3]) and a larger number of demonstrations (typically 50–60). In contrast, our method employs a much simpler model with a maximum of 1.5e+02 parameters and is effective even with a single expert demonstration, as demonstrated in the paper, and is still capable of guaranteeing out-of-sample recovery for the entire state space.
>
>
>
> **References**
>
> [1] Brown, Daniel, et al. "Safe imitation learning via fast Bayesian reward inference from preferences." International Conference on Machine Learning. PMLR, 2020.
>
> [2] Laskey, Michael, et al. "DART: Noise injection for robust imitation learning." Conference on robot learning. PMLR, 2017.
>
> [3] Chi, Cheng, et al. "Diffusion policy: Visuomotor policy learning via action diffusion." The International Journal of Robotics Research, 2023.
>
> [4] Ravichandar, Harish, et al. "Recent advances in robot learning from demonstration." Annual review of control, robotics, and autonomous systems 3.1 (2020): 297-330.

---

### Official Review · Reviewer_AXT4 · 2024-11-04

**Soundness:** 2
**Presentation:** 3
**Contribution:** 2
**Rating:** 5
**Confidence:** 3

**Summary:**

The paper proposes a method (SCDS) for imitating trajectories by learning a contractive latent dynamical systems. Contractivity means, that the resulting (latent) trajectories of any two initial states exponentially converge to the same trajectory with contractivity rate $\gamma$. By enforcing contractivity, the paper aims to put more focus on approaching the goal in the same way as the demonstrations.

For modeling the latent dynamical system, SCDS uses a recurrent equilibrium network (REN, Martinelli et al., 2023), which already enforces contractivity by design. The latent state is mapped to the observed state by linearly projecting it to the desired dimension and then applying a learned bijection. The parameters of the REN, the projection and the bijection are learned end-to-end by minimizing a weighted MSE between the generated trajectory and the different demonstrations, where the trajectories are generated by first obtaining the latent initial state (by applying the inversed bijection of the observed initial state followed by a pseudo-inverse of the linear projection), and then integrating the latent dynamical system using a neural ODE. For computing the MSE between trajectories of potentially different length, SCDS applies dynamic time warping; the weights for the weighted MLE are computed such that the closest demonstrations obtain the largest weights.

Apart from presenting this architecture, the paper shows a lower bound on this weighted MLE for any initial states within a certain region of the demonstrated initial states, and compares the method with baselines on simulated robot environments.

**Strengths:**

The presentation of the method is very clear. Figure 1 illustrates the problem setting quite well, and Figure 2 gives a good overview of the pipeline.

The experimental results are good, and show that the reproduced trajectories behave similar to the demonstration, for initial states that are close to the demonstrated initial states.

The proposed pipeline is novel.

The claims seem to be correct, although the discussion of the experimental results need to be toned down, as the benefits are often not significant (e.g. in Table 1).

**Weaknesses:**

__Relevance__

The proposed method has several limitations:
- First of all, it is to note that the problem setting only involves imitating state trajectories, without accounting for the robot dynamics and therefore the feasibility of the planned motions.
- The methods assumes that all demonstrations end in the same state, which is rarely the case for practical problems.
- By applying DTW, the method is not able to imitate velocity profiles.

__Soundness__

- The weighting for the MLE loss (Eq. 9) is not derived from first-order principles.

- Assumption 4.1 seems to be overly strong, as it assumes that *all* (rather than any) demonstrated initial states are close to $\hat{\mathbf{y}}\_0$.

- The mapping from latent state to observed state is not invertible and therefore requires the use of a pseudo-inverse. As a consequence, the generated trajectory might not start at the correct initial state.

- Although the experiments involve settings with multi-modal initial state distributions, it seems like such settings are not compatible with the aim of learning a dynamical system where all trajectories converge to the same one. I assume that approaching a goal from very directions would only be possible with loose contraction rates, which would defeat the main motivation of the paper.

__Experiments__

- The presentation of the experiments in Table 1 highlight the best mean MSEs, although the confidence intervals are quite large, which is a bit misleading.

- The "OOD" initial states are still rather close to the initial states of the demonstrations. It would be interesting to evaluate the different methods for more distant initial states. It would also be interesting to evaluate the demonstrations at states that appeared during the end of a demonstration.

- Qualitatively, the generated demonstrations tend to collapse to the same trajectory, which is not surprising given the objective. However, oftentimes, we are also interested in capturing the variance of the demonstration, which can be important for variable stiffness control. In that respect, the baseline SDS-EF yields arguably a better imitation.

**Questions:**

- Were the metrics in Table 1 computed by weighting the demonstrations using Eq. 9? Wouldn't this bias the results in favor of the proposed method?

- Why isn't the contractivity in conflict with multimodal trajectory distributions?

---

> ### Author Response · Authors · 2024-11-18
> **Response to reviewer's comments - Part 1**
>
> We are grateful for the reviewer’s constructive and detailed feedback. We have carefully addressed all the comments in the order they were raised and made further refinements to the paper. An updated version of the manuscript is released.
>
> **1. No account for the robot dynamics and therefore the feasibility of the planned motions.**
>
> Our method operates under the “model-free” assumption. For explanation, please refer to the answer to the second question of reviewer hPP6.
>
>
> **2. All demonstrations end in the same state, which is rarely the case for practical problems.**
>
> We have removed the assumption of a fixed final state in the updated version of our paper, as it is unnecessary. A contractive DS converges to a unique target state from any initial condition. While we imposed this characteristic on the data in the original submission, our framework can still be applied to data that does not adhere to this property. In such cases, the model learns an average behavior, converging to a single point rather than replicating each endpoint exactly. An example of this scenario, with demonstrations ending at different points, is illustrated in Figure 13 of the paper, demonstrating that this assumption was not required.
>
> **3. By applying DTW, the method is not able to imitate velocity profiles.**
>
> Our method can imitate any kinematic measurement of the expert’s behavior, including velocity, regardless of the loss function used. This is achieved by augmenting the state to include both position and velocity, hence, doubling the state dimensions.  In Section 5.2, we demonstrated this capability on the LASA-4D and Robomimic-14D datasets, which include both the robot’s joint position and velocity. The results, reported in Table 1, show performance using both MSE and DTW loss functions. Thus, our approach is not limited to imitating position profiles but also learns and reproduces velocity profiles accurately.
>
>
> **4. The weighting for the MLE loss (Equation 9) is not derived from first-order principles.**
>
> While we acknowledge that our weighting method is heuristic, we believe it is sensible as it has the following properties.
> The weighted loss simplifies to the standard average loss for in-sample initial states, as explained in Lines 304-306 of the submitted paper and our answer 5 to reviewer cFmX. Hence, when using the weights in Equation (9), the weighted loss generalizes the average loss.
>
> It provides an intuitive solution for out-of-sample initial states. For initial states different from those in the dataset, it is unclear whether the generated trajectory should be compared to which demonstration. However, it is reasonable to use a weighted combination of the losses with respect to each demonstration, where a higher weight is assigned to the demonstration with a closer initial state. An example depicting this scenario is provided in Figure 3. To our knowledge, previous imitation learning methods have not considered computing out-of-sample loss.
>
>  We hope this clarifies the rationale behind our approach.
>
>
> **5. Assumption 4.1 seems to be overly strong, as it assumes that all (rather than any) demonstrated initial states are close to $\hat{y}_0$.**
>
> Assumption 4.1 is indeed milder than your statement. For example, in the case of two expert demonstrations, Assumption 4.1 requires that the sum of distances from $\hat{y}_0$ to the two demonstrated initial conditions is smaller than $R$. This places $\hat{y}_0$ inside an ellipse with focal points at the two demonstrated initial conditions—a much larger region than requiring all demonstrated initial states to be close to $\hat{y}_0$, as you noted, which corresponds to the intersection of two circles.
>
> We identified and corrected a small typo in Assumption 4.1 in the updated paper. This correction does not affect the proof or the subsequent theorem.
>
> **6. The generated trajectory might not start at the correct initial state, due to using the pseudo-inverse.**
>
> This is correct. When $P_{\boldsymbol{\theta}}$ has full column rank, the pseudoinverse coincides with the left inverse, resulting in $\hat{y}(0) =y_0$, and the generated trajectory starts at the correct initial state. Otherwise, it provides the least-squares approximation to $y_0$.
>
> Based on your comment and comments from other reviewers, we have added this clarification after Equation (6) in the updated paper to highlight the distinction between $\hat{y}(0)$ and $y_0$.

---

> ### Author Response · Authors · 2024-11-18
> **Response to reviewer's comments - Part 2**
>
> **7. Approaching a goal from very directions would only be possible with loose contraction rates, which would defeat the main motivation of the paper.**
>
> We respectfully disagree with the assertion that multi-modal initial state distributions are incompatible with contractivity. Our experiments, such as those shown in Figures 4 and 5, successfully demonstrate the ability of our method to imitate multi-modal demonstrations.
>
> Contractivity ensures that all trajectories converge to the target state from any initial condition, regardless of the contraction rate. In our experiments, trajectories starting from initial conditions within each mode contract together before ultimately converging to the target. This behavior highlights the desired transient dynamics alongside the steady-state guarantees, fulfilling the goals of the paper in both aspects.
>
>
>
>
> **8. Table 1 highlights the best mean MSEs, although the confidence intervals are quite large, which is a bit misleading.**
>
> While the confidence intervals are large in some cases, they are not misleading in selecting the best method. This is because our method achieves the lowest out-of-sample error, which is the most critical criterion for determining the best approach, both in terms of mean and variance. For in-sample errors, there are three cases where the baseline with the lowest mean does not have the lowest variance. To address this, in the final version of the paper, we will mark baselines that are not statistically significantly worse using Welch’s t-test.
>
> The larger confidence intervals in Table 1 arise from the inclusion of multiple tasks with varying levels of difficulty within each dataset. For instance, in the Robomimic dataset, the 'lift' task is relatively easier to imitate than the 'transport' task, leading to larger confidence intervals.
>
>
> **9. Evaluating different methods for more distant initial states and states that appeared during the end of a demonstration**
>
> We highlight that Figure 10 in the original submission analyzes the impact of the spread of initial conditions using a task from the LASA dataset. In the most extreme cases, some initial states are close to the end of the demonstrations. As shown in the figure, SCDS demonstrates effective out-of-sample recovery, even when the initial states are far from the in-sample ones. To increase the visibility of this analysis, we referred to this ablation study at the beginning of Section 5 in the updated paper.
>
> Following your and reviewer hPP6’s comments, we have also included Table 3 in the updated version, comparing SCDS with other baselines when starting from more distant initial states. The results, shown below, indicate that our approach performs significantly better in these scenarios, which is expected given our strong convergence guarantees.
>
> | Expert           | LASA-2D (MSE) | LASA-2D (soft-DTW) | LASA-4D (MSE) | LASA-4D (soft-DTW) |
> |------------------|---------------|--------------------|---------------|--------------------|
> | SNDS            | 4.64 ± 2.50   | 11.06 ± 2.35       | 6.57 ± 3.56   | 14.78 ± 1.08       |
> | BC              | 13.81 ± 6.08  | 29.25 ± 8.45       | 32.73 ± 11.45 | 41.22 ± 11.51      |
> | SDS-EF          | 2.67 ± 0.55   | 12.12 ± 2.29       | 4.19 ± 0.78   | 16.34 ± 3.28       |
> | **SCDS (ours)**        | **0.57 ± 0.23** | **2.12 ± 0.70**    | **1.44 ± 0.35** | **3.40 ± 0.56**    |
> We hope this additional analysis further clarifies the effectiveness of our method in handling distant initial states.
>
> **10. Oftentimes, we are also interested in capturing the variance of the demonstration, which can be important for variable stiffness control. In that respect, the baseline SDS-EF yields arguably a better imitation.**
>
> Our method consistently outperforms SDS-EF in out-of-sample performance across all datasets and achieves comparable or superior in-sample results in our experiments. Notably, we evaluated our method on datasets similar to those used in the SDS-EF paper, including the LASA dataset and the Franka arm. Our experiments also include multi-modal cases, demonstrating that our approach can capture the variance in demonstrations.
> While SDS-EF might perform better in specific scenarios, such as variable stiffness control, this would represent a different problem than the one analyzed in our paper. Additionally, SDS-EF does not claim to specifically address such scenarios. Hence, it is not possible to conclude that SDS-EF yields arguably better results.

---

> ### Author Response · Authors · 2024-11-18
> **Response to reviewer's comments - Part 3**
>
> **11. Were the metrics in Table 1 computed by weighting the demonstrations using Equation 9? Wouldn't this bias the results in favor of the proposed method?**
>
> The weighting method does not bias the results, as explained below.
> For training our approach and evaluating in-sample rollouts, the weighted loss in Equation (8) reduces to an average loss, as noted in Lines 304–306 of the submitted paper. This is the same loss function used in other methods. For evaluating out-of-sample rollouts, the weights in Equation (9) are applied consistently across our method and the baselines.
> Therefore, the metrics in Table 1 are computed consistently for all methods, ensuring no bias in favor of our proposed approach. Additional clarification about the weighting is provided in our response to reviewer cFmX."
>
>
> **12. Why isn't the contractivity in conflict with multimodal trajectory distributions?**
>
> The key factor enabling this behavior is the initial distance between trajectories, $\mathbf{z}_0^a - \mathbf{z}_0^b$, appearing on the right-hand side of the contractivity definition in Equation (2).
>
> Qualitatively, if two trajectories start from nearby initial states, the upper bound on their distance (RHS of Equation (2)) decreases below a specific small threshold $\epsilon$ very quickly, causing them to converge towards the target point together. In contrast, if the initial states are farther apart, it takes longer for the upper bound on their distance to fall below the same $\epsilon$, allowing the trajectories to approach the target from different directions before eventually converging.
>
> Importantly, the generated trajectories from a contractive DS converge to a unique point. This requires a common endpoint for all modes in a multimodal demonstration.
>
> **References**
>
> [1] Khansari-Zadeh, S. Mohammad, and Aude Billard. "Learning stable nonlinear dynamical systems with Gaussian mixture models." IEEE Transactions on Robotics 27.5 (2011): 943-957.
>
> [2] Mirchevska, Branka, et al. "High-level decision making for safe and reasonable autonomous lane changing using reinforcement learning." 2018 21st International Conference on Intelligent Transportation Systems (ITSC). IEEE, 2018.
>
> [3] Bahl, Shikhar, et al. "Neural dynamic policies for end-to-end sensorimotor learning." Advances in Neural Information Processing Systems 33 (2020): 5058-5069.

---

> ### Comment · Reviewer_AXT4 · 2024-11-26
>
> > Assumption 4.1 is indeed milder than your statement. For example, in the case of two expert demonstrations, Assumption 4.1 requires that the sum of distances from $\hat{y}_o$
> to the two demonstrated initial conditions is smaller than $R$. $\hat{y}_o$
> inside an ellipse with focal points at the two demonstrated initial conditions—a much larger region than requiring all demonstrated initial states to be close to $\hat{y}_o$, as you noted, which corresponds to the intersection of two circles.
>
> I don't understand. Consider a 2d-coordinate frame with demonstrations at [-0.5,0] and [0.5,0] and assume $R=1$. The sum of the distances at the position [-1,0], which is on the ellipse and $R$-close to one of the demonstrations is $0.5+1.5>R$. More generally, the sum of distances must always be larger than the maximum distance, and therefore we would need to be close to all demonstrations.
>
>
> > Qualitatively, if two trajectories start from nearby initial states, the upper bound on their distance (RHS of Equation (2)) decreases below a specific small threshold
> very quickly, causing them to converge towards the target point together. In contrast, if the initial states are farther apart, it takes longer for the upper bound on their distance to fall below the same $\epsilon$, allowing the trajectories to approach the target from different directions before eventually converging.
> Importantly, the generated trajectories from a contractive DS converge to a unique point. This requires a common endpoint for all modes in a multimodal demonstration.
>
> But how about two trajectories that start at the same point and split mid-trajectory? E.g. moving around an obstacle from both sides?

---

> ### Author Response · Authors · 2024-11-27
>
> Thank you for taking the time to go through our responses. We address your follow-up questions below. Please let us know if we can clarify further and improve your opinion of our work.
>
> **Consider a 2d-coordinate frame with demonstrations at [-0.5,0] and [0.5,0] and assume $R=1$. The sum of the distances at the position [-1,0], which is on the ellipse and $R$-close to one of the demonstrations is $0.5+1.5>R$. The sum of distances must always be larger than the maximum distance, and therefore we would need to be close to all demonstrations.**
>
> Assumption 4.1 does not state that $\hat{\mathbf{y}}_0$ must be $R$-close to one of the demonstrations. Instead, it requires that the sum of distances to all demonstrations is smaller than $R$, which defines an ellipse.
>
> In your example with demonstrations at $[-0.5, 0]$ and $[0.5, 0]$, the point $[-1, 0]$ is not on the ellipse with $R = 1$. Instead, as you computed, it lies on the ellipse with $R = 2$ ($\vert-1-(-0.5)\vert + \vert-1-0.5\vert = 2 = R$).
>
> To summarize, we consider the set of initial states where the sum of their distances to all demonstrations is smaller than a given $R$. In 2-dimensions with two demonstrations, this corresponds to an ellipse. As you noted, this implies that the distance to each demonstrated initial condition is smaller than $R$.
>
> We believe Assumption 4.1 is not restrictive for two reasons:
>
> * The ellipse can be made arbitrarily large by selecting a larger $R$, though this would loosen the upper bound on the loss. This trade-off is reasonable, as weaker performance guarantees are expected for initial points far from the demonstrations.
>
> * Contractivity and convergence to the target are global properties that hold for all initial states, including those outside the ellipse. The ellipsoidal assumption is only employed for deriving performance bounds.
>
> We hope this clarifies the reasoning behind Assumption 4.1 and addresses your concern.
>
>
> **How about two trajectories that start at the same point and split mid-trajectory? E.g. moving around an obstacle from both sides?**
>
> Since the expert is assumed optimal [1], having diverging situations like the one you described means that the expert behavior is stochastic. Our method focuses on deterministic policies, which are well-suited for deterministic demonstrations. To clarify this, we will explicitly state in the final version of the paper that the initial conditions of expert demonstrations are distinct. To the best of our knowledge, all dynamical system-based imitation learning approaches operate under this assumption (see for example [2, 3]).
>
>
> **References**
>
> [1] Ahmed Hussein et al. Imitation learning: A survey of learning methods. ACM Computing Surveys (CSUR), 2017.
>
> [2] Rana, Muhammad Asif, et al. "Euclideanizing flows: Diffeomorphic reduction for learning stable dynamical systems." Learning for Dynamics and Control (L4DC), 2020.
>
> [3] Hadi Beik Mohammadi et al. Neural contractive dynamical systems. In The Twelfth International Conference on Learning Representations (ICLR), 2024.

---

> > ### Author Response · Authors · 2024-12-02
> >
> > We hope our responses have adequately addressed your comments. However, we remain available for any additional questions or clarifications during these final hours of the discussion period.

---

### Official Review · Reviewer_cFmX · 2024-11-09

**Soundness:** 2
**Presentation:** 3
**Contribution:** 2
**Rating:** 6
**Confidence:** 4

**Summary:**

This paper proposes a method for learning an autonomous dynamical system model from data. Specifically, the data is assumed to be in the form of recorded state trajectories from a dynamical system controlled by an "expert" policy. The proposed model is "contractive" by construction; that is, all solutions of initial value problems (IVPs) using this model exponentially converge towards each other.

The paper claims that using a contractive model improves robustness of the learned model to out-of-sample (OOS) states. Specifically, if the model and expert-controlled systems are initialized at a state in a bounded region near those in the expert-demonstration data, then the error between the model and expert-controlled rolled-out state trajectories differs from the empirical loss by at most a constant. This claim is accompanied by theoretical and simulated experiment results.

**Strengths:**

The paper shows a good effort put into communicating concepts from contraction theory and dynamical systems in an imitation learning context. The simulation experiments encompass a variety of systems that make them potentially compelling, in particular to the robot learning community. The choice of loss term weights in Equation (9) is, to the best of my knowledge, novel; moreover, their use in establishing the bound in Theorem 4.1 is neat.

**Weaknesses:**

My primary issue with this paper is its misuse of the term "policy", and its presentation as a paper that supposedly does policy learning. As mentioned in my summary of this paper, the proposed method learns a model of an _autonomous dynamical system_, i.e., one that is uncontrolled or has a fixed controller. In other words, given a system described by the ODE $\dot{x} = f(x)$, or $\dot{x} = f(x, u) = f(x, \pi(x)) \eqqcolon f_\pi(x)$ with fixed controller $u = \pi(x)$, the method in this paper learns an approximate model $\hat{f}$ for $f$ or $f_\pi$. This is not policy learning, which would instead attempt to explicitly learn a controller $\pi$. Specifically, in an imitation learning context, policy learning would involve trying to learn whatever "expert policy" $\pi$ gives rise to the data in the expert demonstration set.

This is an important distinction to make, particularly for the robot learning community that this paper seems to target. Learning an imitating policy would allow a user to then deploy the policy on the actual robot to autonomously recreate expert demonstrations (within some error). However, just learning a model for the closed-loop "dynamical system + expert policy" does not enable this. This diminishes the proposed contribution.

There are also a couple of mathematical issues (although they may be remediable):
- In the linear projection $P_\theta z$ with $P_\theta \in \mathbb{R}^{N_y \times N_z}$, from the paper and the accompanying code, the authors seem to use a much larger latent dimension $N_z$ than robot state dimension $N_y$. However, $P_\theta$ then cannot be injective (i.e., have full column rank), and so the left inverse $P_\theta^\dagger$ does not exist in general (even if it may still be computable without errors on a computer using floating point arithmetic).
- Each reciprocal in the _harmonic_ mean (not geometric mean as incorrectly mentioned in the appendix on page 16) in Equation (9) is only defined when the initial condition does not match any of the expert trajectories (otherwise you get 1 / 0). However, having the same reciprocal in the numerator may make the limit tend to 1 (since $\lim_{a \to 0^+} (1/a) / (1/a + 1/b) = \lim_{a \to 0^+} 1 / (1 + a/b) = 1)$, although this should be clarified in the paper. More importantly, the submitted code does not seem to use these weights at all and instead just uses the regular mean (see `loss = criterion(out, expert_trajectory).mean()` on line 83 in `ren_trainer.py`).

**Questions:**

- Please clarify whether the work does or does not learn an implementable controller / policy for the system (e.g., robot). In the case of the latter, are there "real-world scenarios" (page 6, line 293) where this might be useful, particularly for the robot learning community? Describing these would be crucial to better establishing the proposed contribution.

- Can you remedy the issue of non-injective $P_\theta$ in Equation (7)? Since there are multiple possible $z_0 \in \mathbb{R}^{N_z}$ corresponding to a given $\hat{y}_0 \in \mathbb{R}^{N_y}$ when $N_z > N_y$, is there a way to make a "best" choice?

- Please clarify where the submitted experiment code actually uses the weights $ \lambda_m(\hat{y}_0) $. Do they make a difference in the results when compared to just using the standard mean?

---

> ### Author Response · Authors · 2024-11-18
> **Response to reviewer's comments**
>
> Thank you for your thoughtful and detailed comments. We have responded to your feedback point by point, as originally outlined, and have also implemented further improvements to the paper. An updated version of the manuscript is released.
>
> **1. Misuse of the term "policy"**
>
> We agree that there is a slight misuse of the term 'policy.' In our paper, the policy mapping is split into two steps: the DS in Equation (4) maps the current state to a desired future state, and then a low-level controller calculates the action needed to reach this target state. This process, as illustrated in Figure 15, means that the DS and low-level controller together constitute the policy (there was a typo in Figure 15 in the original submission, please refer to the updated version).
>
> To address this, we have added further clarifications in Section 2.3 of the updated paper explaining how the DS and low-level controller form the policy and highlighted that we refer to the DS alone as the 'policy' for simplicity.
>
>
> **2. Does the work learn an implementable controller/policy for the system?**
>
> Yes. Indeed, we implemented our method on the Franka arm and Jackal mobile robots using an off-the-shelf low-level controller in the Isaac Lab simulator. The only required step for implementing our controller is designing the low-level controller which maps the states planned by the DS in our policy to motor commands. As discussed in our initial response to reviewer AXT4, designing such controllers is a field of independent research. Furthermore, this reliance on a low-level controller is common in imitation learning [1] and reinforcement learning [2] methods that operate at the level of high-level actions (e.g., joint positions) rather than low-level (e.g., torque) commands. Thus, this reliance is not specific to our method.
> Additionally, the sim-to-real gap for controllers that output the joint position, such as ours, is significantly smaller than those outputting lower-level actions [3]. Therefore, we expect a smooth transition to real-world applications.
>
>
> **3. The left inverse does not exist in general.**
>
> We do not use the left inverse but rather the Moore–Penrose pseudoinverse, which exists for any matrix and provides a generalized solution in cases where a left inverse does not exist.
>
> When $P_{\boldsymbol{\theta}}$ has full column rank, the pseudoinverse coincides with the left inverse, resulting in $\hat{y}(0) = y_0$. Otherwise, it provides the least-squares approximation to $y_0$ [4].
>
>
> Based on your comment, we have added this clarification after Equation (6) in the updated paper to highlight the distinction between $\hat{y}(0)$ and $y_0$.
>
>
> **4. Equation (9) is the harmonic mean, not the geometric mean, as mentioned on page 16.**
>
> Page 16 applies the inequality of arithmetic and geometric means (AM-GM inequality) to establish a relationship, but it does not state that Equation (9) is a geometric mean. Based on your feedback, we have extended the proof of Theorem 4.1 in the updated paper to clarify how the AM-GM inequality is employed.
>
>
> **5. Where the submitted experiment code uses the weights $\lambda_m(\hat{y}_0)$?**
>
> We clarify how the weights $\lambda_m(\hat{y}_0)$ are used in training and evaluation stages:
>
> * Training: The average loss is used during training. As you mentioned, the weights $\lambda_m(\hat{y}_0)$ are binary (0 or 1) when the initial state matches a dataset sample. Consequently, as noted in Lines 304–306 of the submitted paper, the weighted loss simplifies to an average loss. Using only the initial conditions in the dataset ensures fairness when comparing our method to other baselines, as it prevents our approach from utilizing additional initial condition samples.
>
> * In-sample rollout evaluation (yellow rows, Table 1): In this case, the weights remain binary and the loss again reduces to an average loss.
>
> * Out-of-Sample rollout evaluation (green rows, Table 1): For out-of-sample rollouts, where the initial conditions differ from those in the dataset, the weights are no longer binary and are adjusted accordingly. In this case, we use the weighted loss to evaluate our algorithm as well as the baselines.
> For further details on why the weights are necessary for out-of-sample rollouts, please refer to our response to reviewer AXT4.
>
> **References**
>
> [1]  Chi, Cheng, et al. "Diffusion policy: Visuomotor policy learning via action diffusion." The International Journal of Robotics Research, 2023.
>
> [2] Bahl, Shikhar, et al. "Neural dynamic policies for end-to-end sensorimotor learning." Advances in Neural Information Processing Systems, 2020.
>
> [3] Salvato, Erica, et al. "Crossing the reality gap: A survey on sim-to-real transferability of robot controllers in reinforcement learning." IEEE Access 9 (2021): 153171-153187.
>
> [4] Penrose, E. T. (1956). Towards a theory of industrial concentration.

---

> > ### Comment · Reviewer_cFmX · 2024-11-25
> >
> > Thank you for the clarifications! Indeed, the use of a contractive system as a "motion planner" in conjunction with a low-level controller, altogether as the policy makes more sense. There is still the issue of possibly infeasible state trajectories mentioned by another reviewer, but these are generally less of a problem with over-actuated robot systems like manipulators with >= 7 DOF.
> >
> > Regarding the loss function weights, I see now why the code just uses the mean then. It would be interesting to see if there is a test-time benefit to training with additional initial conditions outside of those in the expert dataset. I would not call it "unfair" if your method can take advantage of this; it would present an interesting ablation and might even increase the potential contribution of the paper.
> >
> > The mathematical issues I mentioned are less critical than the main contribution to policy learning (which you have clarified). I will nitpick that while the pseudo-inverse always exists, you will in general not be able to recover a unique solution to $Pz = y$ (where I've dropped the bijective maps $g$ for simplicity) if $P$ is not invertible. In particular, if $P \in \mathbb{R}^{N_y \times N_z}$ is wide ($N_z > N_y$), then at best you can find a minimizer for $\min_z \lVert Pz - y\rVert$ which is only unique if $P$ is injective, although there is a unique minimizer $z$ with minimum norm $\lVert z \rVert$ (see Theorem 3.1 in "Regression and the Moore-Penrose Pseudoinverse" by A. Albert). Indeed, contraction properties generally only transfer through smooth diffeomorphisms (i.e., smooth, differentiable bijections); as soon as you start changing the dimension of the system, maintaining contraction is not as straightfoward.
> >
> > Given the clarifications made by the authors, I have raised my score. I think there is a possibly interesting contribution made and I would not be strongly opposed to accepting the paper; however, I think the paper could benefit from a bit of polish and more careful consideration of the use of the weights in training.

---

> > > ### Author Response · Authors · 2024-11-25
> > >
> > > Thank you! Regarding your comment on the transfer of contraction properties only through smooth diffeomorphisms, we highlight **Proposition 2.1** which states that **any linear transformation, which entails changing the dimension of the system, preserves contraction**. The proof is provided in Appendix A.1.
> > >
> > > We really appreciate your careful consideration of our response and for raising your score. We also acknowledge your suggestion to further polish the paper and will thoughtfully address the use of weights in training in the final revision.

---

### Official Review · Reviewer_NYvK · 2024-11-10

**Soundness:** 3
**Presentation:** 3
**Contribution:** 3
**Rating:** 6
**Confidence:** 4

**Summary:**

The paper addresses limitations in traditional imitation learning (IL) by introducing a framework based on contractive dynamical systems. The goal is to create imitation policies that reliably generalize to out-of-sample (OOS) states and remain stable under environmental perturbations. The approach leverages recurrent equilibrium networks (RENs) and coupling layers to ensure contractivity across all parameter values. By formulating the imitation problem as a differentiable ODE, the authors achieve efficient training with unconstrained optimization. The framework is demonstrated on robotics tasks, with improved OOS performance compared to existing methods.

**Strengths:**

- The paper introduces a novel approach to imitation learning by employing contractive dynamical systems, going beyond traditional stability guarantees to enhance policy reliability in out-of-sample (OOS) regions and under perturbations.

- The authors provide theoretical bounds for loss term.

- The idea of integrating recurrent equilibrium networks (RENs) and coupling layers to ensure that the policy is contractive regardless of parameter choices seems interesting. This idea enables efficient training with unconstrained optimization and simplifying policy deployment.

- Experimental results on LASA and Robomimic datasets demonstrate improvements in OOS recovery and imitation accuracy over baselines.

- I appreciate that the authors provided code base for reproducibility. Also, the paper is well-written.

**Weaknesses:**

- No comparison of computational complexity along with wall clock time against the baselines is provided.

- In the policy representation, the usage of RENs and coupling layers adds complexity. I am expecting a discussion around this as well as the complexity of the baselines.

- While I appreciate that the authors provided bounds on the loss term, the result is too simple, I was expecting some sort of sample complexity result to actually see the sample requirements for their method.

- SNDS paper also includes discussion on high-dimensional Snake demonstrations. I don't see experiments on such data in this work.

- I have some questions that I would like to get answers to in the Questions section.

**Questions:**

- The paper assumes that there is a given final state and that the initial states may be perturbed. However, how to deal with the cases where the target state is not fixed? For example, in the MuJoCo environments, the only requirement is to have the robot work successfully for 1000 steps and there is no constraint on final state as such. How to deal with this case?

- The above also brings me to the question that if for every expert trajectory in the dataset, the final state changes. Will your method work in this case? If not, how would you address that? Currently, I see that the number of steps in each trajectory can be different but the final state is still the same.

- How much is the algorithm sensitive to the quality of expert demonstration? If the data comes from a suboptimal policy, what should one expect would happen in this framework?

- Maybe I missed this but does the method do interaction with the environment during learning? If yes, then some comparison of number of interactions made by different algorithms is needed.

- The model’s performance is sensitive to the contraction rate, as mentioned by the author, I am curious how critical this is when we have to run the algorithm on some new task.

---

> ### Author Response · Authors · 2024-11-18
> **Response to reviewer's comments - Part 1**
>
> We appreciate the reviewer’s insightful feedback and have addressed each comment in the order it was presented. Furthermore, we made additional refinements to enhance the paper’s quality according to the comments. An updated version of the manuscript is released.
>
> **1. Computational time was not provided.**
>
> The computational times of our method and other baselines for different datasets are detailed in Appendix E6. To increase visibility, we have now included a direct reference to these results at the beginning of Section 5 in the main body of the updated paper.
> Our results indicate that, when training all baselines for the same number of epochs, our method is less computationally efficient in lower-dimensional datasets but is comparable in high-dimensional ones. However, this comparison is conservative, as our approach typically requires fewer epochs to achieve similar performance, resulting in a shorter actual training time.
> Importantly, our method offers a stronger contractivity guarantee, compared to only a stability guarantee in other baselines and no guarantees in naive behavioral cloning, which we believe offsets the time trade-off.
>
>
> **2. Using RENs and coupling layers increases the complexity. How does it compare with the baselines?**
> While RENs and coupling layers increase the complexity of our policy representation, they bring two key advantages: (1) high flexibility and (2) guaranteed contractivity for any parameter choice, eliminating the need for contractivity constraints during training (see Lines 58–64).
>
> A comparison with baselines using simpler policy representations is provided in Lines 65–74. These simpler baselines lack inherent contractivity and therefore require constrained optimization for training, which is computationally intensive. To mitigate this, they assume specific policy classes, which reduces expressiveness.
>
> The only baseline that, like ours, inherently guarantees contractivity through its policy representation is the approach by [1]. However, as noted in Lines 75–81, their policy structure is not necessarily simpler than ours.
>
> **3. The generalization bounds must contain sample complexity results to reflect the sample requirements for the method.**
>
> Our bounds in Theorem 4.1 and Corollary 4.1.1 inherently reflect sample complexity through the parameter $M$, which denotes the number of expert demonstrations. For a fixed $R$, representing a fixed level of uncertainty in the initial conditions, the upper bounds in Theorem 4.1 and Corollary 4.1.1 become tighter as $M$ increases. This aligns with the expected behavior of improved generalization with more data.
>
> In the extreme case where $M \rightarrow \infty$, the conservatism due to the uncertainty in the initial state (captured by term (ii) in Theorem 4.1 and Corollary 4.1.1) vanishes. This demonstrates that the bounds are non-vacuous.
>
> **4. Experiments on the Snake Environment employed in the SNDS paper.**
>
> We have tested our approach on the Messy Snake dataset following the same experimental setup as the SNDS paper and reported the results in Appendix E.7 of the updated version. As shown in Figure 15, our method (SCDS)  generates contracting trajectories that better mimic the expert data compared to SNDS. Quantitatively, SCDS achieves an average DTW error of 0.074, significantly outperforming SNDS, which has an average DTW error of 0.158.
>
>
> **5. The paper assumes that there is a given final state.**
>
> We have removed the assumption of a fixed final state in the updated version of our paper, as it is not required. This assumption was inspired by the LASA dataset, in which all trajectories reach a given final state, but is not integral to our framework.
> A contractive DS converges to a unique target state from any initial condition. While we imposed this characteristic on the data in the original submission, our framework can still be applied to data that does not adhere to this property. In such cases, the model learns an average behavior, converging to a single point rather than replicating each endpoint exactly. An example of this scenario, with demonstrations ending at different points, is illustrated in Figure 13 of the paper, demonstrating that this assumption was not required.
>
> **6. Sensitivity to the quality of expert demonstration.**
>
> As is typical in imitation learning, our framework assumes that the expert demonstrations represent an optimal policy. This assumption is foundational in imitation learning, as discussed on page 6 of [5]: 'In imitation learning, the demonstrations provide the optimal action to a given state, and so the agent learns to reproduce this behavior in similar situations.' Addressing cases with suboptimal demonstrations is beyond the scope of this work.

---

> ### Author Response · Authors · 2024-11-18
> **Response to reviewer's comments - Part 2**
>
> **7. Does the method interact with the environment during learning?**
>
> No, our method does not interact with the environment during learning. Instead, it relies on an offline dataset, as introduced in Equation (1) of the paper.
>
>
> **8. How critical is the sensitivity to the contraction rate when we have to run the algorithm on some new task?**
>
> Choosing the contraction rate is not a critical issue since it can be learned from data, as detailed in Appendix E2, allowing it to adapt to different tasks as needed. Our experiments in Figure 11 show the efficacy of learning the contraction rate.
>
> **References**
>
> [1] Hadi Beik Mohammadi, Søren Hauberg, Georgios Arvanitidis, Nadia Figueroa, Gerhard Neumann, and Leonel Rozo. Neural contractive dynamical systems. In The Twelfth International Conference on Learning Representations, 2024.
>
> [2] Harish Ravichandar, Athanasios S Polydoros, Sonia Chernova, and Aude Billard. Recent advances in robot learning from demonstration. Annual review of control, robotics, and autonomous systems, 2020.
>
> [3] Amin Abyaneh, Mariana Sosa Guzm ́an, and Hsiu-Chin Lin. Globally stable neural imitation policies. In IEEE International Conference on Robotics and Automation (ICRA), 2024.
>
> [4] S Mohammad Khansari-Zadeh and Aude Billard. Learning stable nonlinear dynamical systems with Gaussian mixture models. IEEE Transactions on Robotics, 2011.
>
> [5] Ahmed Hussein, Mohamed Medhat Gaber, Eyad Elyan, and Chrisina Jayne. Imitation learning: A survey of learning methods. ACM Computing Surveys (CSUR), 2017.

---

> > ### Comment · Reviewer_NYvK · 2024-11-26
> >
> > Thank you for addressing my concerns. After reviewing the changes, I am happy to increase the score.

---

> > > ### Author Response · Authors · 2024-11-26
> > >
> > > Thank you for taking the time to carefully review our response and raising the score. We truly appreciate the opportunity to address your concerns. If there are any additional points or clarifications that could further strengthen your perspective on our paper, please do not hesitate to let us know.

---

### Author Response · Authors · 2024-11-19
**Link to the revised manuscript**

As promised earlier, we are pleased to share the updated version of our text, revised after incorporating feedback received during the rebuttal phase. We have made every effort to reflect the reviewers' comments, including conducting additional experiments detailed in the appendix and updates to the main text.

You can access the updated document through the following link: [Link to the revised manuscript](https://drive.google.com/file/d/1vJXn50jwaPQqxjGAoQSvFguyVykPR_mv/view?usp=sharing). The same version is also uploaded through OpenReview.

Thank you for your insightful feedback and continued support.
Should you require any further clarification, please do not hesitate to reach out.

---

### Author Response · Authors · 2024-11-25
**We're here for any questions**

As the rebuttal period is coming to an end, we wanted to express our gratitude for your insightful feedback—it has greatly improved our paper. We hope our clarifications have addressed your concerns and are happy to answer any remaining questions or issues during these final days.

---

### Meta-Review · Area_Chair_gBQa · 2024-12-21

**Metareview:**

This paper introduces SCDS, a novel imitation learning framework that leverages contractive dynamical systems to improve out-of-sample (OOS) generalization and robustness to perturbations. SCDS employs recurrent equilibrium networks (RENs) and coupling layers to enforce contractivity. SCDS is evaluated on various robotics tasks, demonstrating improved OOS performance compared to existing baselines. It also makes a theoretical contribution as a derived upper bound on the loss for initial states within a multi-focal ellipse region.

The reviewers generally agree on the originality and clarity of the paper, highlighting the novel integration of contractive RENs, neural ODE solvers, and coupling layers. By using contractive dynamical systems, the paper can provide theoretical guarantees of the SCDS method, although not all reviewers agree that this is an important or impressive result. The paper also includes empirical results in simulated robot environments, where the method outperforms three baselines. After the rebuttal period, the paper’s clarity was improved.

The main weaknesses of this paper relate to the assumptions about the experimental setting. Both theoretically and empirically, the paper investigates only a limited scope for what “out-of-sample” means, and the provided theoretical guarantees have to make somewhat unrealistic assumptions about the initial state relative to the expert trajectories. The method is also more than 2x slower to train relative to the baselines, and is only really more useful in the “out-of-sample” setting (for in-sample, it performs comparably or sometimes worse than baselines).

Nevertheless, all reviewers raised their scores during the rebuttal period, and my own careful reading of the paper suggests that the authors were very diligent in their experiments (with a very thorough appendix) and have scoped the main claims of the paper in a manner appropriate with the results. The combination of theoretical investigation and empirical validation, in a domain for which people care about, will also make this a relevant submission for a sizeable chunk of the ICLR audience. Therefore, I recommend acceptance.

**Additional Comments On Reviewer Discussion:**

One reviewer brought up concerns about computational runtimes. This was added to the appendix, and highlighted in the main text during the rebuttal period.

One reviewer asked for results on the Snake environment used in one of the baselines (SNDS), which the authors provided during the rebuttal. In this environment, the SCDS method also outperforms SNDS.

Multiple reviewers mentioned concerns about how the weights for the loss are determined. The authors clarified what the weights are during training, and admitted that while the weighted choice was heuristic, it makes practical sense (I agree).

Multiple reviewers brought up concerns and questions with the assumptions and proofs. These were generally well addressed by the authors, and contributed to the increased scores during the rebuttal period from the reviewers.

---

### Decision · Program_Chairs · 2025-01-22

Accept (Poster)